# StressID: a Multimodal Dataset for Stress Identification

**Hava Chaptoukaev**[1]    **Valeriya Strizhkova**[2]    **Michele Panariello**[1]    **Bianca D'Alpaos**[1]
**Aglind Reka**[2]    **Valeria Manera**[3]    **Susanne Thümmler**[3]
**Esma Ismailova**[3,4]    **Nicholas Evans**[1]    **François Bremond**[2,3]
**Massimiliano Todisco**[1]    **Maria A. Zuluaga**[1*]    **Laura M. Ferrari**[2,5*]

[1]EURECOM, Sophia Antipolis, France
[2]Inria, Valbonne, France
[3]CoBTek, Université Cote d'Azur, Nice, France
[4] Mines Saint-Etienne, Centre CMP, Gardanne, France
[5]The Biorobotics Institute, Scuola Superiore Sant'Anna, Pontedera, Italy
`{chaptouk,panariel,evans,todisco,zuluaga}@eurecom.fr`
`{valeriya.strizhkova,francois.bremond,laura.ferrari}@inria.fr`

## Abstract

`StressID` is a new dataset specifically designed for stress identification from unimodal and multimodal data. It contains videos of facial expressions, audio recordings, and physiological signals. The video and audio recordings are acquired using an RGB camera with an integrated microphone. The physiological data is composed of electrocardiography (ECG), electrodermal activity (EDA), and respiration signals that are recorded and monitored using a wearable device. This experimental setup ensures a synchronized and high-quality multimodal data collection. Different stress-inducing stimuli, such as emotional video clips, cognitive tasks including mathematical or comprehension exercises, and public speaking scenarios, are designed to trigger a diverse range of emotional responses. The final dataset consists of recordings from 65 participants who performed 11 tasks, as well as their ratings of perceived relaxation, stress, arousal, and valence levels. `StressID` is one of the largest datasets for stress identification that features three different sources of data and varied classes of stimuli, representing more than 39 hours of annotated data in total. `StressID` offers baseline models for stress classification including a cleaning, feature extraction, and classification phase for each modality. Additionally, we provide multimodal predictive models combining video, audio, and physiological inputs. The data and the code for the baselines are available at `https://project.inria.fr/stressid/`.

## 1   Introduction

While a healthy amount of stress is necessary for functioning in daily life, it can rapidly begin to negatively impact health and productivity when it exceeds an individual's coping level. Negative stress can be a triggering or aggravating factor for many diseases and pathological conditions [16], and frequent and intense exposures to stress can cause structural changes in the brain with long-term effects on the nervous system [10]. Monitoring of stress levels could play a major role in the prevention of stress-related issues, and early stress detection is vital in patients exhibiting emotional disorders, or in high-risk jobs such as surgeons, pilots or long-distance drivers.

---

*Equal contribution.

37th Conference on Neural Information Processing Systems (NeurIPS 2023) Track on Datasets and Benchmarks.

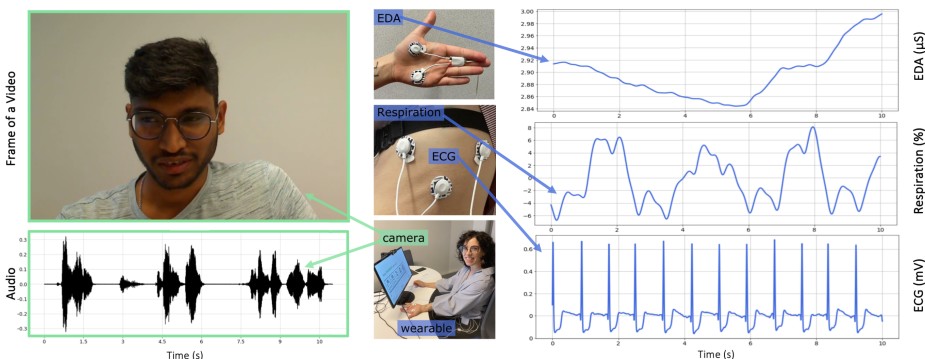

Figure 1: Data collection set-up of `StressID`.

In the last few years, machine and deep learning have been playing major roles in stress recognition research. In practice however, building robust and reliable models for stress identification requires; 1) understanding and integrating the differences between subgroups of the population to ensure bias free applications, 2) integrating the relationships between physical and physiological responses to stress, 3) studying responses to various categories of stressors, as the perception of stress differs strongly from one individual to another. An essential element to such analyses is high-quality and versatile multimodal datasets that include varied categories of stressors, and are recorded on large and diverse populations. However, existing datasets do not answer these needs. They are generally restricted in size (i.e. a few dozen of participants) and a majority is focused on a single source of data (i.e. physiological signals, video or audio) – although multimodal datasets have considerable advantages [26, 29]. Moreover, existing datasets often provide imbalanced subject responses, due to both an inability of the recording protocol to elicit strong reactions and a lack of diversity in the stimuli – making it difficult to deploy deriving analyses to real-life applications.

To address these limitations, we propose `StressID`, a novel multimodal dataset with facial video, audio, and physiological data. To the best of our knowledge, `StressID` is one of the largest available multimodal dataset in the field that includes varied stimuli. It is composed of 65 subjects and more than 39 hours of annotated data in total. `StressID` is designed specifically for the identification of stress from different triggers, by using a guided breathing task, 2 video clips, 7 different interactive *stressors*, and a relaxation task. As illustrated in Figure 1, `StressID` uses a collection of wearable sensors to record the physiological responses of the participants, namely, an Electrocardiogram (ECG), an Electrodermal Activity (EDA) sensor, and a respiration sensor. The data is coupled with synchronized facial video and audio recordings. Each task is associated with 6 different annotations: 4 scores from a self-assessment rating perceived stress and relaxation, along with valence and arousal based on the Self-Assessment Manikin (SAM) [9]; and 2 discrete labels derived from the 4 self-assessments. These data annotations serve to train supervised models.

We summarize our main contributions as follows:

- A novel multimodal dataset focused on stress-inducing tasks, composed of ECG, EDA, respiration, facial video, and audio recordings. The modalities are synchronized and annotated with self-assessments from the participants evaluating their levels of relaxation, stress, valence, and arousal.

- An easy to reproduce experimental protocol for recording behavioral and physiological responses to diverse triggers, using wearable and global sensors.

- Instructions for using the presented dataset and an open-source implementation of several baseline models for stress recognition from video, audio, and physiological signals respectively, as well as multimodal models combining the three inputs.

The remainder of this paper is organized as follows. Section 2 provides an overview of the existing datasets for stress recognition. Section 3 describes the dataset design and its contents. In Sections 4 we present multiple baselines for stress detection using machine learning. In Section 5 and 6, we discuss the limitations and ethical considerations of our work. Finally, we summarize our work and discuss future directions.

## 2  Related work

Table 1 places `StressID` in the context of related stress recognition datasets. The SUS datasets [52] gather the recordings of 35 subjects collected during aircraft communication training. This unimodal collection of datasets only features audio recordings without self-assessments or external annotation and employs an uncommon elicitation task. SADVAW [55] is a dataset composed of 1236 video clips from 41 Korean movies, making the setting closer to the real world and including a broader range of responses. However, it features video recordings exclusively, restricting deriving applications to computer vision systems only. Among the works investigating the physiological aspect of stress, DriveDB [25] collects physiological data from 9 subjects exposed to driving-related tasks. The lack of self-assessment or external annotations significantly limits the accuracy of measuring stress. In addition, the dataset is collected in the very specific setting of driving, with a narrow range of stressors – considerably restricting its usage. WeSAD [48] and CLAS [36], two of the most widely explored datasets for stress recognition, contain physiological data from 15 and 62 subjects respectively, collected using wearable devices. The participants partake in various tasks, combining perceptive stressors in the form of audiovisual stimuli, with several variations of the Trier Social Stress Test (TSST) [3]. However, they do not include any behavioral modalities.

There exist a few multimodal datasets for stress recognition, such as MuSE [27] and SWELL-KW [33]. They feature a broader set of modalities and are collected in laboratory environments imitating real-life activities. MuSE participants are elicited through audiovisual and public speaking tasks. SWELL-KW participants perform office work on several topics designed to elicit different emotions. These datasets are limited in size with recordings of respectively 28 and 25 subjects. Finally, the distracted driving dataset [54] gathers recordings of 68 subjects in the setting of simulated driving with stress-inducing distractions. The lack of diversity in the stimuli restricts subsequent applications to the setting of driving. Moreover, cardiac activity is acquired in terms of heart rate, which does not allow the extraction of heart rate variability (HRV) measures, a key measure in the identification of stress [30].

**Comparison with the state-of-the-art.** `StressID` aims to fill the gap in the existing related work. It features both physiological and behavioral modalities, includes a large number of participants, exploits varied stimuli, and includes participants' replies to 4 self-assessment questions providing insights on the subject's emotional state. Although CLAS [36] and WeSAD [48] present similar experimental set-ups, they focus on physiological modalities and do not include behavioral data. Instead, `StressID` features three types of modalities: video, audio, and physiological signals capturing complementary information. While MUSE [27] and SWELL-KW [33] are also multimodal datasets recorded in similar conditions, they are very limited in size. On the contrary, with 65 subjects recorded `StressID` is one of the largest datasets designed for stress identification. Finally, although the size and modalities of the distracted driving dataset [54] are comparable to `StressID`, it relies on very environment-specific stressors, whereas `StressID` includes emotional video-clips, cognitive tasks, and social stressors based on public speaking, which represents a key aspect to guarantee the collection of a wide range of responses. To summarize, `StressID` is the first multimodal dataset for stress identification that is recorded on a large number of participants but also features a wide range of stimuli ensuring more versatility in deriving applications.

Table 1: Comparison of `StressID` to related datasets.

| Dataset | #Subjects | Modalities | Stressors | Data annotations |
|---|---|---|---|---|
| SUS | 35 | Speech | Aircraft communication training | Stressor-based |
| SADVAW | - | Video | - | External annotations |
| DriveDB | 9 | EMG, EDA, ECG, HR, Respiration | Driving tasks | Stressor-based |
| WeSAD | 15 | ECG, EDA, EMG, BVP, Respiration, Temperature, Acceleration | TSST, Audiovisual | Stressor-based, PANAS [56], STAI [51], SAM [9] |
| CLAS | 62 | ECG, PPG, EDA, Acceleration | Cognitive load, Audiovisual | SAM |
| MuSE | 28 | EDA, HR, Breath rate, Temperature, Face and upper body video, Audio | Public speaking, Audiovisual | PSS [34], SAM, External annotations |
| SWELL-KW | 25 | ECG, EDA, Face and upper body video, Posture, Computer logging | Office work with interruptions and time pressure | NASA task load [24], SAM, Stress assessment |
| Distracted Driving dataset | 68 | EDA, HR signal, Respiration, Face video, Driving performances | Simulated driving with distractions | Stressor-based, NASA task load, SAM |
| StressID | **65** | **EDA, ECG, Respiration, Face video, Speech** | **Cognitive load, Public speaking, Audiovisual** | **SAM, Stress assessment** |

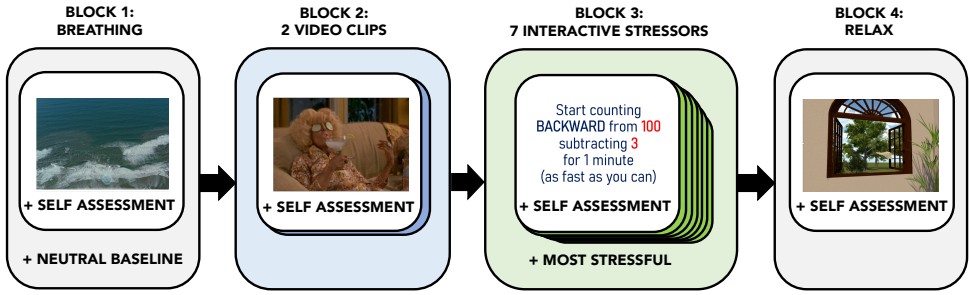

Figure 2: Overview of the experimental protocol. The experiment consists of 11 tasks divided into four blocks: a guided breathing task, 2 emotional video clips, 7 interactive stressors, and a relaxation task.

## 3 StressID Dataset

`StressID` takes a step towards building more robust and reliable applications for automated stress identification by enabling the design of versatile and bias-free algorithms. To achieve this, `StressID` features responses to several categories of stress-inducing stimuli to account for the variability of responses from one individual to another, rather than focusing on a single task. In addition, the large number of participants of `StressID` enables analyses of the demographics associated with stress, based on factors such as gender or age, thus advancing towards reducing representation bias by integrating these differences in subsequent algorithms. Ultimately, the design of the `StressID` dataset supports a variety of learning pipelines, by offering possibilities for the analysis of subject-specific, task-specific and modality-specific responses to stress. We describe the design of `StressID` in Section 3.1. We then introduce the resulting dataset in Section 3.2, and outline our data annotation process in Section 3.3.

### 3.1 Dataset Design

#### 3.1.1 Experimental Protocol

Figure 2 illustrates the experimental protocol used to collect `StressID`. It consists of 11 tasks separated by self-assessments and grouped into 4 blocks: guided breathing, watching emotional video clips, a sequence of interactive tasks, and a relaxation phase. Tasks have been designed to elicit 3 different categories of responses; 1) stimulate the audiovisual cortex of the participants, 2) increase the cognitive load by soliciting attention, comprehension, mental arithmetic or multi-tasking abilities, and 3) elicit psycho-social stress leveraging on public speaking as a stressor. All stimuli are easy to implement and do not require any special setup [5]. The full instructions given to participants are provided in Appendix D.2.

**Guided breathing.** The first block of the protocol consists of the single task of *Breathing*. The participants watch a guided breathing video of 3 minutes. It aims to relax and reset to neutral the emotional state of the subjects. This recording is used as a baseline for the non-verbal neutral state of each participant. After the breathing task, the participants count forward for 1 minute.

**Emotional video-clips.** This block consists in watching 2 emotional videos clips, retrieved from the FilmStim database [47]. These videos have been selected to elicit specific emotional responses.

- *Video1 :* an extract from the movie *There's something about Mary*, selected to elicit low arousal and positive valence in the participants.

- *Video2 :* an extract from the movie *Indiana Jones and the Last Crusade*, selected to elicit high arousal and negative valence.

**Interactive tasks.** This block consists of a sequence of 7 interactive stressors based on well-established clinical methods to induce stress [7]. All the tasks have a strict requirement for response in 1 minute and the order of the stressors is designed to be unexpected to the participants.

- *Counting1 :* a Mental Arithmetic Task (MAT) designed to increase the participants' cognitive load through arithmetic operations with a varying range of difficulty. In this task, the participants receive the instructions to count backwards from 100 subtracting 3 as fast as they can.

- *Counting2 :* another MAT of increased difficulty. Participants are asked to count backward from 1011 subtracting 7 as fast as they can.

- *Stroop :* a variant of the Stroop Color-Word Test [53], selected to increase the cognitive load by soliciting the attention and reactivity of the participants.

- *Speaking :* a Social Evaluative Task (SET), leveraging public speaking as a social stressor. The subjects are instructed to explain their strengths and weaknesses, emulating stressful job interview conditions.

- *Math :* a task designed to increase the mental workload. The participants are asked to resolve 20 mathematical problems in one minute.

- *Reading :* a task composed of 2 phases and designed as a TSST variation. Participants have to read a text, in the first step, and then explain what they read, in the next step, thus simultaneously soliciting comprehension abilities and using speaking as a stressor.

- *Counting3 :* a MAT with added difficulty. Participants are instructed to count backwards from 1152 subtracting 3, as fast as they can, while repeating an independent hand movement. This task is designed to increase the mental workload by soliciting participants' multi-tasking abilities.

At the end of the third block, the participants are asked to designate the task perceived as most stressful.

**Relaxation.** The last block of the experimental protocol is solely composed of the *Relax* task. It consists of a 2 minute and 30 seconds long relaxation part, where participants are instructed to watch a relaxing video [23].

### 3.1.2 Sensors

Three different physiological signals are collected in `StressID`: electrocardiogram (ECG), electrodermal activity (EDA), and respiration signal. They are recorded using the BioSignalsPlux acquisition system[*]. The BioSignalPlux kit consists of a 4-channel hub communicating via Bluetooth with the OpenSignals (R)evolution platform for data visualization and acquisition, connected to an ECG, EDA, and a respiration sensor. The hub ensures the synchronized recording of up to 4 sensors simultaneously. The ECG is acquired with 3 Ag/AgCl electrodes located on the ribs of the non-dominant side of the subjects. The EDA is measured with 2 Ag/AgCl electrodes attached to the palm of the non-dominant hand. The respiration is measured through a chest belt with an integrated piezoelectric sensing element. The selected devices have a high signal-to-noise ratio [42, 43, 44], and all physiological signals are acquired with a sampling rate of 500 Hz and resolution of 16 bits per sample.

The video and audio are acquired using a Logitech QuickCam Pro 9000 RGB camera with an integrated microphone. The video is acquired with a 720p resolution and a rate of 15 frames per second. The audio is recorded at a sampling rate of 32kHz and a resolution of 16 bits per sample.

## 3.2 Dataset Description

### 3.2.1 Recruitment and Recording

Most of the participants of `StressID` are Science, Technology, Engineering, and Mathematics (STEM) students and workers. In total, 65 healthy participants were recruited on a voluntary basis, without compensation. They included 18 women and 47 men of ages ranging between 21 and 55 years old (29y.o. $\pm$ 7). Among the participants, 32% were master students and interns, 20% PhD students, and the remaining 48% represented diverse tertiary professions. All subjects were required to have sufficient proficiency in English and they were requested to sign a consent form to participate.

---

[*]biosignalsplux, PLUX wireless biosignals S.A. (Lisbon, Portugal)

The participants could either consent to, **Option A:** research use and public release of all their recorded data, including identifying data (i.e. physiological, audio, and video). **Option B:** research use of all their recorded data, but no public release of identifying data (i.e. only physiological and audio data, but no video). Among the 65 participants, 62 opted for option A and 3 opted for option B (2 women and 1 man).

Each participant was recorded in a single session, lasting approximately 35 minutes. They were instructed not to smoke, intake caffeine, or exercise 3 hours before the experiment. At the beginning of each session, they were introduced to the purpose and content of the study. The experiments are conducted entirely in English. The experimental protocol was identical for all participants, and the experimenter was always present in the room during the recording.

### 3.2.2 Dataset Composition

Following data collection, we split each recorded session into individual tasks: one 3 minutes breathing recording (block 1), 2 recordings corresponding to the watching of the video clips of respectively 2 and 3 minutes (block 2), 7 separate 1-minute recordings of the interactive tasks (block 3), and a 2 minute and 30 seconds long relaxation recording (block 4). As the guided breathing, the video clips and the relaxation parts do not carry meaningful audio, the audio part of the dataset consists of the 7 talking tasks only. During the acquisitions, due to camera malfunctions, 8 video and audio recordings were damaged. More information about the available tasks for each modality can be found in Appendix A.1. After splitting, `StressID` is composed of 711 distinct annotated recordings of the physiological modalities, 587 annotated videos, and 385 annotated audio recordings. In total, the final task-split dataset amounts to approximately 19 hours of annotated physiological data, 15 hours of annotated video data, and 6 hours of annotated audio data, thus amounting to more than 39 hours of data in total. Each task is identified in the dataset by `subjectname_task`, where the task names are as described in Section 3.1.1. This convention facilitates different types of analyses, whether subject-specific or task-specific.

### 3.3 Data Annotation

Each task is annotated using the answers to self-assessment questions. The first 2 questions establish the participants' perceived stress and relaxation levels on a 0-10 scale. Additionally, they answer the SAM [9] to assess their valence and arousal on a 0-10 scale. Research suggests relaxation and stress conditions can be described in different quadrants of the arousal-valence space. For instance, high arousal and negative valence are characteristics of emotional stress induced by threatening stimuli [15], while low arousal and positive valence are characteristics of a calm and relaxed state [37].

The distributions of the `StressID` self-assessments are reported in Figure 3. The analysis of the distributions highlights a positive correlation between stress and arousal, as well as relax and valence. This suggests that across subjects and tasks, a high arousal is associated with a higher level of stress, and a positive valence corresponds to a higher level of relaxation. In addition, the marginal distributions of stress and relax ratings (Figure 3) highlight a balance in the perceived stress and relaxation levels of the participants across the whole experiment, suggesting that the experimental protocol of `StressID` can arouse proportional instances of stress and relaxation. Furthermore, the distribution of arousal is significantly skewed towards a high rating across the dataset, while valence is centered around a neutral value, highlighting the ability of the protocol to create a high involvement in the participants and elicit strong responses. An extended analysis of the self-assessment distributions analyses can be found in Appendix E.

We propose 2 discrete labels that can be used to train supervised models: a 2-class label and a 3-class one. The 2-class label is computed using the stress self-assessment of each task by splitting the 0-10 scale at 5. Precisely, tasks with self-assessment of stress below 5 are considered **not stressed** (0) while tasks with self-assessment equal or above 5 are **stressed** (1). The 3-class label is based on the results outlined by [15, 37], which are in line with the observations drawn from Figure 3. It allows the prediction of **relaxed** vs. **neutral** vs. **stressed**. We considered a subject to be **relaxed** (0) for a task where they reported a valence rating above 5, an arousal rating below 5, and a perceived relaxation rating above 5. Similarly, we label tasks with arousal levels above 5, valence levels below 5, and perceived stress levels above 5 as **stressed** (2), and **neutral** (1) otherwise.

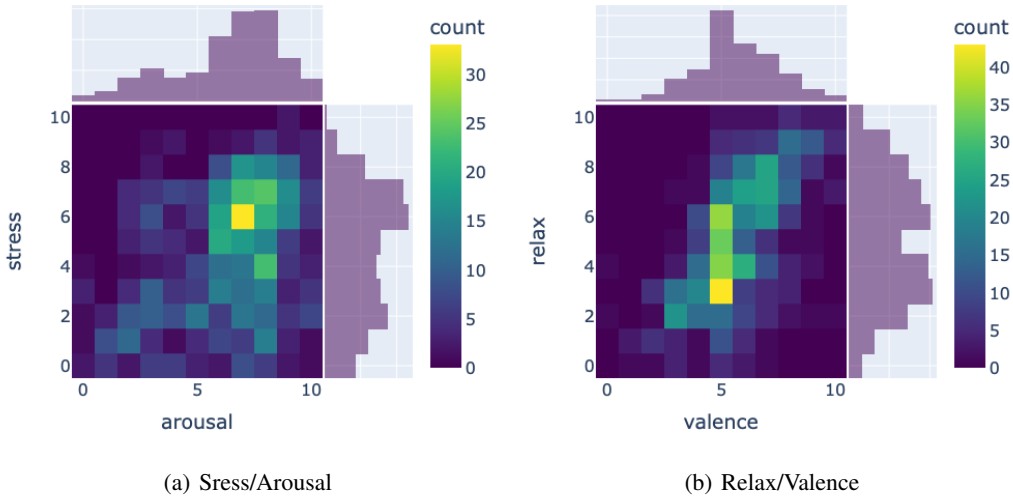

(a) Sress/Arousal  (b) Relax/Valence

Figure 3: Distribution of the self-assessment answers. (left) Joint and marginal distributions of stress and arousal. (right) Joint and marginal distributions of the relax and valence ratings.

# 4    Baselines

We implement several unimodal and multimodal baselines combining features extracted from video, audio, and physiological inputs. We train the models to perform 2-class classification, i.e. binary discrimination between stressed and not stressed, as well as 3-class classification. In all the experiments, we generate 10 random splits, using 80% of the tasks for training, and 20% for testing for each split. The results are averaged over the 10 repetitions. To ensure robustness to potential imbalance resulting from the train-test splits, the results are assessed using the weighted F1-score and the balanced accuracy on the test data. The full list of extracted features, additional experiments, model hyperparameters, and training details are reported in Appendix F. The implementation of all the baselines can be found at `https://github.com/robustml-eurecom/stressID`.

## 4.1    Unimodal Baselines

Each unimodal baseline is trained and tested on all available tasks of the corresponding modality, i.e. 715, 587, and 385 tasks respectively for the physiological, video, and audio baseline. In the following, we describe the baselines for each modality. The obtained results are reported in Table 2.

**Physiological Signals.** In line with the literature on stress recognition from physiological signals [5, 20, 21], we propose a baseline including pre-processing of the signals, feature extraction, and classification. In a first step, the ECG, EDA, and respiration signals are filtered with Butterworth filters to reduce high-frequency noise and baseline wander. Then, 35 ECG features, 23 EDA, and 40 respiration features are extracted. These include HRV features in the time domain including the number of R to R intervals (RR) per minute, the standard deviation of all NN intervals (SDNN), the percentage of successive RR intervals that differ by more than 20ms and 50ms (pNN20 and pNN50), or the root mean square of successive RR interval differences (RMSSD), as well as frequency-domain, and non-linear HRV measures. We have extracted statistical features of the Skin Conductance Level (SCL) and Skin Conductance Response (SCR) components of the EDA, including the slope and dynamic range of the SCL, along with time domain features including the number of SCR peaks per minute, the average amplitude of the peaks, and average duration of SCR responses. In addition, we have extracted Respiration Rate Variability (RRV) features in the time and frequency domains. The resulting handcrafted (HC) features are then classified using classical Machine Learning (ML) algorithms: a Random Forests (RF) classifier, Support Vector Machines (SVM), and a Multi-Layer Perceptron (MLP) with hyperparameters chosen by Cross-Validation (CV).

Table 2: Performances of unimodal baselines for the classification of stress. Each baseline is trained and tested on all available tasks of the corresponding modality.

| Baseline | 2-class | | 3-class | |
|---|---|---|---|---|
| | F1-score | Accuracy | F1-score | Accuracy |
| Physio. HC features + RF | **0.73 ± 0.02** | **0.72 ± 0.03** | 0.55 ± 0.04 | 0.56 ± 0.03 |
| Physio. HC features + SVM | 0.71 ± 0.02 | 0.71 ± 0.02 | **0.59 ± 0.04** | **0.59 ± 0.03** |
| Physio. HC features + MLP | 0.70 ± 0.03 | 0.70 ± 0.03 | 0.54 ± 0.04 | 0.53 ± 0.04 |
| AUs + kNN | 0.70 ± 0.04 | 0.69 ± 0.04 | 0.54 ± 0.05 | 0.53 ± 0.05 |
| AUs + SVM | 0.69 ± 0.04 | 0.69 ± 0.04 | 0.55 ± 0.05 | 0.54 ± 0.04 |
| AUs + MLP | **0.70 ± 0.03** | **0.70 ± 0.03** | **0.55 ± 0.03** | **0.55 ± 0.03** |
| Audio HC features + kNN | 0.67 ± 0.06 | 0.60 ± 0.05 | 0.53 ± 0.04 | 0.52 ± 0.04 |
| Audio HC features + SVM | 0.61 ± 0.06 | 0.54 ± 0.03 | 0.53 ± 0.08 | 0.48 ± 0.04 |
| W2V 2.0 classifier | **0.70 ± 0.02** | **0.66 ± 0.03** | **0.56 ± 0.04** | **0.52 ± 0.04** |

**Video Data.** We propose a baseline employing Action Units (AU) and eye gaze for the classification of stress. AUs are commonly used as features in stress recognition applications [22, 27, 2]. They are fine-grained facial muscle movements [18], each relating to a subset of extracted facial landmarks [40]. Each AU is described in two ways: presence, if the AU is visible in the face, and intensity, indicating how intense the AU is on a 5-point scale (minimal to maximal). After downsampling the recordings to 5 frames per second, we use the OpenFace library [8] to extract eye gaze and AUs from each video frame. We extract the following AUs: 1, 2, 4, 5, 6, 7, 9, 10, 12, 14, 15, 17, 20, 23, 25, 26, 28, and 45. As eye gaze features, we use two gaze direction vectors computed individually for each eye by detected pupil and eye location. The averages and standard deviations of each AU and eye gaze directions are computed across time frames. The resulting 84-component vector is used as input to several models: a k-Nearest Neighbors (kNN) algorithm, an SVM, and an MLP with 4 layers of width 256. In line with [27], the number of layers and layer width of the MLP are chosen by CV in {2,3,4} and {64, 128, 256} respectively. We use ReLU activation and the MLP is trained for 100 epochs with cross-entropy loss optimized using Adam [31] with an initial learning rate of $1e-3$.

**Audio Data.** We propose two baselines for speech signals: the first employs HC features and ML algorithms, and the second is built on the Wav2Vec 2.0 (W2V) model [49, 6]. Both techniques involve downsampling from the original 32 kHz audio to 16 kHz, and the application of amplitude-based voice activity detection (VAD) [32] prior to feature extraction to eliminate non-speech segments. The first baseline relies on a plethora of specific audio features [46, 4] widely used in the literature on emotion recognition from speech [1, 5, 35]. These include Mel Frequency Cepstral Coefficients (MFCCs) and their first and second derivatives, which characterize the short-term power spectrum and its dynamics. The spectral centroid, bandwidth, contrast, flatness, and roll-off, which together provide a rich statistical representation of the spectral shape. Harmonic and percussive components are also extracted, with tonal centroid features being computed for the harmonic component. The zero-crossing rate is a simple measure of the rate of sign changes; the rate of zero-crossings relates directly to the fundamental frequency of the speech signal. Last, we include tempogram ratio features [39] which represent local rhythmic information. We compute the mean and standard deviation over time for all features, thereby resulting in feature vectors for each, which are then concatenated to form a comprehensive feature vector of 140 components, and used as input for ML algorithms.

The second baseline employs a large, pre-trained W2V model. The W2V 2.0 model produces features capturing a wealth of information relevant to diverse tasks including emotion recognition [11, 50, 14]. Features are extracted every 20 ms and averaged over time to obtain a single 513-component embedding per utterance, and are then classified using a linear classification layer optimized with Adam, cross-entropy loss, and an initial learning rate of $1e-3$, until convergence.

### 4.2 Multimodal Baselines

Multimodal baselines that combine the features extracted from all 3 sources are evaluated on the tasks that feature all modalities only, i.e. 370 tasks, to avoid learning with severely missing values. This subset of `StressID` is composed of talking tasks exclusively, i.e. all tasks without the audio modality are excluded. In this setting, the dataset presents a strong imbalance in the labels (70%

Table 3: Performances of multimodal baselines for the classification of stress, compared to unimodal models. All baselines are trained and tested only on tasks featuring all modalities, i.e. 370 tasks.

| Baseline | 2-class | | 3-class | |
|---|---|---|---|---|
| | F1-score | Accuracy | F1-score | Accuracy |
| Physiological only | $0.66 \pm 0.05$ | $0.58 \pm 0.04$ | $0.50 \pm 0.05$ | $0.48 \pm 0.06$ |
| Video only | $0.67 \pm 0.03$ | $0.62 \pm 0.04$ | $0.58 \pm 0.05$ | $0.56 \pm 0.05$ |
| Audio only | $0.67 \pm 0.04$ | $0.62 \pm 0.04$ | $0.56 \pm 0.06$ | $0.54 \pm 0.06$ |
| Feature fusion + SVM | $0.64 \pm 0.09$ | $0.56 \pm 0.05$ | $0.55 \pm 0.06$ | $0.51 \pm 0.05$ |
| Feature fusion + MLP | $0.66 \pm 0.04$ | $0.61 \pm 0.03$ | $0.51 \pm 0.07$ | $0.51 \pm 0.07$ |
| Feature fusion + DBN | $0.58 \pm 0.06$ | $0.52 \pm 0.05$ | $0.30 \pm 0.09$ | $0.32 \pm 0.04$ |
| SVM + Sum rule fusion | $\mathbf{0.72 \pm 0.05}$ | $0.64 \pm 0.05$ | $0.62 \pm 0.05$ | $\mathbf{0.58 \pm 0.07}$ |
| SVM + Product rule fusion | $0.71 \pm 0.05$ | $0.63 \pm 0.05$ | $0.61 \pm 0.05$ | $0.56 \pm 0.07$ |
| **SVM + Average rule fusion** | $\mathbf{0.72 \pm 0.05}$ | $\mathbf{0.65 \pm 0.05}$ | $\mathbf{0.63 \pm 0.05}$ | $\mathbf{0.58 \pm 0.07}$ |
| SVM + Maximum rule fusion | $\mathbf{0.72 \pm 0.05}$ | $0.64 \pm 0.05$ | $0.61 \pm 0.06$ | $0.57 \pm 0.07$ |

stress). We use Minority Over-sampling Techniques (SMOTE) [13] to balance the training set in each of the 10 repetitions, and leave the test sets untouched.

We propose fusion models combining all features using the most prominent fusion methods in the literature: feature-level and decision-level fusion [1, 38]. For **feature-level fusion**, unimodal HC features are combined into a single high-dimensional feature vector, used as input for learning algorithms. Similarly to [28, 12], we evaluate feature-level fusion combined with SVM, MLP classifiers, and Deep Belief Networks (DBN). For **decision-level fusion**, following [57, 45], we train independent SVMs for each modality using the HC features as input, and integrate the results of the individual classifiers at the decision level, i.e. the results are combined into a single decision using ensemble rules. The results for all multimodal baselines for the 2-class and 3-class classification are reported in Table 3. To ensure fairness in the comparison, the multimodal baselines are evaluated against best-performing HC and ML-based unimodal baselines (Section 4.1), trained on the subset of the 370 tasks featuring all modalities. Additional results for all other modality combinations are reported in Appendix F.2.2.

## 5 Limitations

First, this dataset is recorded in a controlled environment specifically designed to elicit responses. Experiments conducted in laboratory settings do not take into consideration the external factors that contribute to the psychological mental state of participants and typically assume a stress reaction is an isolated occurrence. In reality, human emotions are complex and are influenced by combinations of factors. In addition, the process of attaching electrodes to the participants may be stressful in itself. Therefore, the signals recorded in this setting are not necessarily representative of real-life situations. In consequence, although models built on the `StressID` dataset can learn to reliably recognize a response to stress-inducing stimuli, the discrimination between positive and negative, or short-term and long-term stress is a more sensitive task. Second, relying on self-assessed scales for data annotation is a participant-subjective process, and can lead to bias in subsequent analyses. Perception of stress and relaxation can vary a lot from one participant to another. Nevertheless, analyses described in Section 3.3 highlight a coherent distribution of the self-reported annotations across participants and the whole experiment. Third, although all participants recruited for the study are proficient in English, the act of speaking English itself can be stress-inducing for non-native speakers. Fourth, the audio component of the dataset suffers from an uneven distribution of labels, as the verbal tasks are associated with higher levels of stress. Fifth, `StressID` suffers from missing modalities for some participants. Finally, `StressID` presents a gender imbalance representative of the female/male ratio in STEM studies and workforce [19]. This is a limitation `StressID` shares with competitor datasets [33, 54, 48, 36, 27], and a common issue in human data collection, in general [17, 41]. Additional experiments sensitising users to the effect of gender bias and demonstrating how `StressID` can effectively be used to build equitable applications, are reported in Appendix F.2.3.

# 6 Ethical Considerations and Dataset Accessibility

The recording and usage of human activity data are associated with ethical considerations. The `StressID` project is approved by the ethical committee of the Université Cote d'Azur (CER). The experiments have been conducted under agreement n° 2021-033 for data collection, and n° 2023-016 for the publication of the dataset. The participants explicitly consent to the recording of their session, the dataset creation, and its release for research purposes following General Data Protection Rules (GDPR). The personal information (sex, age, education), and the acquired physiological and audio signals are pseudonymized, and an alphanumeric code is given for each participant. Video data can not be anonymized and is treated as sensitive data.

Given the identifying nature of the facial videos, the dataset is made accessible through open credentialized access only, for research purposes. Users are required to sign an end-user license agreement to request the data. Once validated a secured set of credentials is granted to access to the dataset. The dataset uses a proprietary license for research purposes and it is hosted on Inria servers using storage intended for long-term availability. The code uses an open-source license. We are aware that despite all precautions, the dataset can be misused by bad-intentioned users. The data and the code for the baselines are available at `https://project.inria.fr/stressid/`.

Lastly, systems that use the dataset for modeling and understanding the mechanisms of human stress conditions need to be aware of the potential imbalance in representation in the dataset. Participants for the data collection were included in our dataset without restrictions on gender, race, age, or education level – instead favoring sample size.

# 7 Conclusion

We present `StressID`, a dataset for stress identification featuring three categories of data modalities and three different types of stimuli. The experimental protocol designed to collect the `StressID` dataset is easy to replicate and can be adapted to additional sensors or stressors. The equipment used for the data collection is affordable, and the selected devices guarantee low noise in the recordings.

The multimodal nature of `StressID` offers a large set of possible uses cases and applications. On one hand, diverse modalities carry complementary information that can be jointly exploited: video and audio capture the behavioural component of emotions, the reactions that are visible from outside, while the physiological signals capture valuable internal states not visible on camera such as cardiac activity, or skin sweating. By providing access to multiple synchronized modalities, `StressID` enables cross-modal analyses that have the potential to improve the understanding of the relationships between video, audio, and physiological responses to stress. On the other hand, the dataset design also offers the possibility to develop models of different natures, by focusing on a single modality. Moreover, it allows a wide range of applications, including subject-specific, or task-specific studies.

`StressID` dataset can contribute to advance research in multiple fields. First, it has the potential to improve the understanding of the sources, demographics, and both physical and physiological mechanisms of stress responses. It is designed for the development of reliable algorithms for stress identification that can improve the quality of life of our society by helping prevent stress-related issues. For instance, early stress recognition can be beneficial for people suffering from neurological or developmental disorders with emotion deregulation, such as autism, for whom the increase of stress can cause disruptive behaviors. Second, `StressID` can help improve affect understanding, as it offers the possibility to analyze and understand the correlation patterns between the distributions of perceived stress and emotion, how these correlations relate to different categories of stimuli, or how they impact subsequent stress and emotion recognition algorithms. Finally, `StressID` is useful to the machine learning and deep learning communities as well, as it can be used to further evolve multimodal learning algorithms, to develop strategies for learning with unevenly represented modalities, or to study how to make algorithms learning with human data more reliable.

To foster reproducibility, `StressID` also offers a set of baseline experiments. Although the proposed models focus on predicting two discrete labels designed to illustrate the predictive potential of our dataset, they represent a good starting point for future work, to which researchers and developers can benchmark their work. In this context, a natural extension of this work would be the implementation of a web service that tracks and centralizes the performances of models developed using `StressID`.

## Acknowledgments and Disclosure of Funding

HC, VS, FB and MAZ are supported by the French government, through the 3IA Côte d'Azur Investments in the Future project managed by the National Research Agency (ANR) (ANR-19-P3IA-0002). MP, NE and MT are supported by the ANR RESPECT Project (ANR-18-CE92-0024). LMF has been partially supported by the Ville de Nice and the French government, through the UCAJEDI Investments in the Future project managed by the ANR (ANR-15-IDEX-01) and by PNRR- Investment 1.5 Ecosystems of Innovation, Project Tuscany Health Ecosystem (THE), Spoke 3 "Advanced technologies, methods, materials and heath analytics" CUP: I53C22000780001.

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
