# StressID: a Multimodal Dataset for Stress Identification

**Hava Chaptoukaev**[1]    **Valeriya Strizhkova** [2]    **Michele Panariello**[1]    **Bianca D'Alpaos**[1]
**Aglind Reka**[2]    **Valeria Manera**[3]    **Susanne Thümmler**[3]
**Esma Ismailova**[3,4]    **Nicholas Evans**[1]    **François Bremond**[2,3]
**Massimiliano Todisco**[1]    **Maria A. Zuluaga**[1*]    **Laura M. Ferrari**[2,5*]
[1]EURECOM, Sophia Antipolis, France
[2]Inria, Valbonne, France
[3]CoBTek, Université Cote d'Azur, Nice, France
[4] Mines Saint-Etienne, Centre CMP, Gardanne, France
[5]The Biorobotics Institute, Scuola Superiore Sant'Anna, Pontedera, Italy
`{chaptouk,panariel,evans,todisco,zuluaga}@eurecom.fr`
`{valeriya.strizhkova,francois.bremond,laura.ferrari}@inria.fr`

## Supplementary Materials

---

*Equal contribution.

# A Overview of Dataset Contents

The `StressID` dataset contains high-resolution synchronized data from a suite of wearable sensors and global sensors. The dataset provides answers to 4 self-assessment questions in terms of perceived stress, relaxation, arousal, and valence, for each task. These annotations can be used to create robust labels for supervised learning. Fig. 1 presents a summary of the dataset contents and size. The remainder of the supplementary materials provide all additional information about the `StressID` project.

## A.1 Dataset Size

`StressID` contains recordings of 65 subjects. Among the 65 participants, 62 agreed to the public release of all their data and 3 opted for the release of non-identifying data only, i.e. physiological and audio. During the acquisitions, due to anomalous camera malfunction, the video and/or audio recordings of 9 participants – including 1 participant who chose to not share the video data, were damaged. Consequently, the `StressID` dataset contains physiological recordings for 65 participants, video recordings for 54 participants, and audio for 55 participants.

Following data collection, each recorded session is split into individual tasks: one 3-minutes breathing recording, 2 recordings corresponding to the watching of the emotional video clips of respectively 3 and 2 minutes, 7 separate 1-minute recordings of interactive tasks, and a relaxation recording of 2 minutes and 30 seconds, resulting to up to 11 tasks per subject for the physiological and video modalities, and 7 tasks per subject for the audio component composed of verbal-tasks only. Besides the 9 recordings that do not have video or audio, 6 individual task recordings are removed from the public dataset due to technical issues during the execution of the task.

Ultimately, the entire `StressID` dataset consists of 711 tasks for the physiological data, 587 tasks for video data, and 385 tasks for audio data. In total, it represents approximately 1119 minutes of physiological signals recordings, 918 minutes of video recordings, and 385 minutes of audio. Table 1 summarizes the number of instances and total duration of the annotated tasks across the 65 participants, in each modality. More information about missing modalities or missing tasks is provided on the technical file available on the `StressID` webpage[*].

Table 1: Counts and durations of each tasks, in each modality.

| Task/Stressor | Count physiological (min) | Count video (min) | Count Audio (min) |
|---|---|---|---|
| Breathing | 65 (195) | 52 (156) | 0 (0) |
| Video1 | 64 (185) | 52 (150) | 0 (0) |
| Video2 | 64 (126) | 53 (104) | 0 (0) |
| Counting1 | 65 (65) | 54 (54) | 55 (55) |
| Counting2 | 65 (65) | 54 (54) | 55 (55) |
| Stroop | 65 (65) | 54 (54) | 55 (55) |
| Speaking | 65 (65) | 54 (54) | 55 (55) |
| Math | 65 (65) | 54 (54) | 55 (55) |
| Reading | 65 (65) | 54 (54) | 55 (55) |
| Counting3 | 65 (65) | 54 (54) | 55 (55) |
| Relax | 63 (158) | 52 (130) | 0 (0) |
| **Total** | **711 (1119)** | **587 (918)** | **385 (385)** |

---

[*]`https://project.inria.fr/stressid/dataset-composition-details/`

# StressID Dataset Facts

**Dataset** StressID

## Motivation

**Summary** A multimodal dataset for stress identification from video, speech and physiological data from wearable sensors.
**Example Use Cases** Stress identification, emotion recognition, task classification
**Original Authors** H. Chaptoukaev, V. Strizhkova, M. Panariello, B. D'Alpaos, A. Reka, V. Manera, S. Thümmler, E. Ismailova, N. Evans, F. Bremond, M. Todisco, M. A. Zuluaga, L. M. Ferrari

## Metadata

| | |
|---|---|
| **URL** | `https://project.inria.fr/stressid/` |
| **Keywords** | Stress recognition, multimodal, wearable sensors |
| **Format** | .csv, .txt, .mp4, .wav |
| **Ethical review** | Approved by CER/CERNI |
| **Licence** | Proprietary |
| **First release** | 2023 |

## Sensors

| | |
|---|---|
| **ECG** | BioSignalsPlux ECG sensor |
| **EDA** | BioSignalsPlux EDA sensor |
| **Respiration** | BioSignalsPlux Piezoelectric chest-belt |
| **RGB Camera** | Logitech QuickCam Pro 9000 RGB |
| **Audio** | QuickCam Pro 9000 integrated microphone |

## Data Annotations

| | |
|---|---|
| **Self-assessments** | Stress, relax, arousal, valence |
| **Labels** for supervised learning | Binary stress, 3-class stress |

## Annotated Tasks

| | |
|---|---|
| **Relaxing** | Guided breathing, relaxation |
| **Audiovisual** | Video clips |
| **Interactive stressors** | Cognitive tasks, public speaking, multi-tasking |

## Participants

| | |
|---|---|
| **Count** | 65 |
| **Gender** | 72%Male, 28%Female |
| **Age** | 29 ± 7 years |
| **Background** | 32%Master students, 20%PhD students, 48%Tertiary |

## Dataset Size

| | |
|---|---|
| **Total size** | 5.29GB |
| **Physiological** total duration across subjects and across tasks | 1119 min |
| **Video** total duration across subjects and across tasks | 918 min |
| **Audio** total duration across subjects and across tasks | 385 min |

Figure 1: A dataset summary card for `StressID`, constructed based on [2, 5].

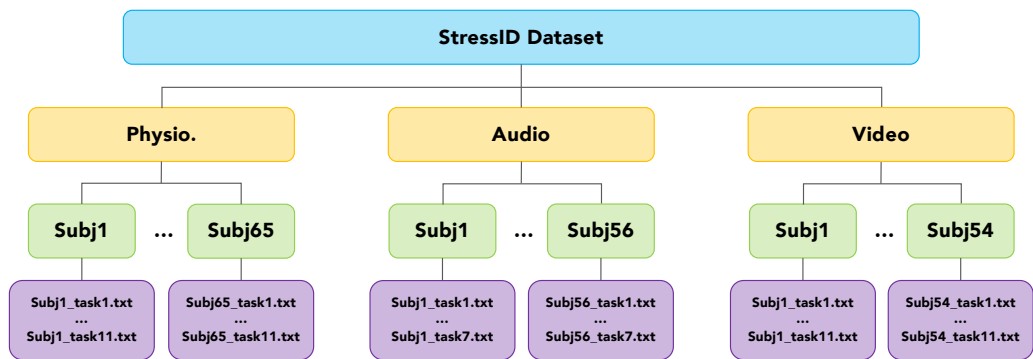

Figure 2: Organisation of the `StressID` Dataset repository. The dataset is grouped by modality. In each modality repository, the tasks are grouped by subject in separate repositories.

## A.2 Data Formats and Organization

The organization of the dataset at download is illustrated in Fig. 2. Each individual task is identified in the dataset by `subjectname_task`. For each modality, all tasks are grouped by subject into separate repositories. Additionally, we provide 3 `.csv` files containing self-assessments and labels; the `self_assessments.csv` file gathers perceived levels of relaxation, stress, arousal and valence for each subject and each task, `labels.txt` corresponds to the labels used to compute the `StressID` baselines, and `labels_supplementary.txt` provides additional labels used in supplementary experiments described in F.

For each task, data from all wearable sensors is organized into a single `.txt` file, making it easily usable with any programming language including Python, Matlab, or C++. Each file contains 3 synchronized entries corresponding to the ECG, EDA, and respiration data respectively. All the physiological signals are sampled at 500Hz with a resolution of 16 bits per sample.

In a similar fashion, for each task, the video data from the Logitech QuickCam Pro 9000 RGB camera is contained in a `.mp4` generated video file. All videos are acquired at a 720p resolution and sampled at 5 frames per second. Audio data is recorded at 32kHz with 16 bits per sample. After cutting and processing, the signals are downsampled to 16kHz. The audio for each task is included in the dataset as an uncompressed `.wav` file.

## B Dataset Publishing and Usage

### B.1 Dataset Hosting and Licensing

The dataset and code are available for researchers. The dataset uses a custom non-commercial proprietary license for research purposes only. It is made accessible through credentialized access. Users are required to sign an end-user license agreement to request the data. Once validated, a link to the repository with a username and a password will be given to grant access. The `StressID` dataset represents 5.29 GB of data. It is hosted on Inria servers, using storage intended for long-term availability, and ensuring sufficient space to hold all collected data. This space is maintained by the INRIA infrastructure team. It is also easily accessible to the research team, allowing new data to be added as it is collected, or withdrawn if needed. This storage thus, allows the dataset to be both dynamic and persistent. The front-end website[*] describes the StressID project, access instructions for downloading the data, the adopted sensors, the recording framework, dataset composition details, and the baseline models. It is hosted on Inria servers intended for long-term persistent websites and also maintained by the infrastructure team. The website acts as a portal pointing to all relevant visualizations, data, code, and instructions. The code for the baselines and analyses uses an open-source 3-Clause BSD License [6], and is available on GitHub[*]. It includes `ReadMe` files describing the

---

[*] `https://project.inria.fr/stressid/`
[*] `https://github.com/robustml-eurecom/stressID`

code structure, installation, and usage. In addition, third-party services for archival code repositories will be explored.

## B.2   Intended Uses and Ethical Considerations

StressID is conceived to further develop research on automated stress recognition. The dataset is a resource of annotated synchronized physiological signals, videos, and audio data, captured while subjects are involved in tasks specifically designed to elicit stress reactions. Various use cases include extracting characteristics of stress from each modality, analyzing correlations between various modalities, analyzing how the modalities relate to specific tasks, training learning pipelines for the identification of stress in diverse verbal and non-verbal tasks, and training pipelines to discriminate between audiovisual stimuli, stressors designed to increase the cognitive load or stressors based on public speaking.

The dataset is made available for research purposes. All personal information about the participants including, sex, age, and background of participants, although not published, is pseudonymized. The acquired physiological and audio signals are also pseudonymized, while video data can not be anonymized. Although the participants explicitly consent to the recording of their session, the dataset creation, and its public release for research purposes, no attempts should be made to actively identify the subjects included in the dataset. The data should also not be modified or augmented in a way that further exposes the subjects' identities.

In general, recording and usage of human activity data is associated with high ethical implications, including privacy, bias, and impact on society. If new projects use the StressID experimental protocol to replicate the study, using similar sensors and identifying modalities, the privacy of any new subjects should be protected, and the implications of the project clearly described to the participants. In addition, future applications that use the StressID protocol and/or dataset for building and training new learning pipelines, should consider the societal implications of their work. StressID is designed as a resource for improving the monitoring, modeling, and understanding of the mechanisms of human stress conditions. All intended applications have the potential to improve the quality of life of the population by helping prevent stress-related issues. However, researchers need to be aware of potential representation bias in their analyses. Indeed, StressID and subsequent analysis may present an imbalance in gender, race, age, or background of the participants – which could lead to unanticipated consequences. Additional information is provided about the participants' demographics along with the dataset and should be taken into account when developing new applications based on the StressID dataset.

We are aware that despite all the precautions, the dataset can be misused by bad-intentioned users. The authors declare that they bear all responsibility in case of any violation of rights during the collection of the data or other work, and will take appropriate action when needed, e.g., by removing data with such issues.

## C   Human Subjects Considerations

The StressID project was approved by the Institutional Review Board (IRB) of Université Côte d'Azur, namely the Committee on Ethics for Non-Interventional Research (CERNI/CER). The project has been conducted under agreement n° 2021-033 for data collection, and n° 2023-016 for the publication of the dataset. Subjects were recruited by email, and word of mouth primarily. They are composed of 32% Master students, 20% PhD students. The other 48% represent tertiary professions. Before the start of the experiment, they were introduced to the purpose and contents of the project, and public release modalities and privacy concerns were described. Participants signed a recording consent form and a media release form. Each subject participated on a voluntary basis. Each experiment lasted approximately 50 minutes including preparing sensors, calibration, and the 35 minutes long experiment.

Safety risks include those associated with the wearable sensors used in StressID. Notably, the use of Ag/AgCl electrodes can cause discomfort or cutaneous irritations in subjects – however, using clinical grade electrodes during the data collection campaign, we did not encounter any issue of this type. In addition, the wearable devices used in StressID should not be used in patients with implanted electronic devices of any kind, including pacemakers, electronic infusion pumps,

stimulators, defibrillators, or similar. All subjects are made aware of this fact, and cannot participate in the experiment if they fall in any of the mentioned categories. The experiment presented no safety risks associated with tasks. Participants were informed they could stop the experiment at any time.

Given the identifying nature of the videos, privacy is a primary concern in this project. Therefore, the data collection protocol of `StressID` considered the privacy risks for the participants as much as possible. The goals and implications of publishing personally identifiable facial videos were clearly described to each participant, and a dedicated media release consent form was signed to acknowledge participants' willingness for their video to be part of the public release of the data. Ultimately, participants could select between two options: **Option A:** research use and public release of all their recorded data, including identifying data (i.e. facial videos), and **Option B:** research use of all their recorded data, but no public release of identifying data. The videos of the participants who selected option B are removed from the public version of the dataset.

The subjects are also informed that they can withdraw their consent at any time. In that case, the data collected prior to the creation of the database will be destroyed. If the database has already been created and the subjects have given consent to the use of physiological data or audio, as these are pseudo-anonymous, they cannot be deleted. Video data will not be shared with other people after the withdrawal request. However, data that has already been shared cannot be modified. Once the database has been shared with other authorized researchers, the subjects will no longer be able to exercise their right of withdrawal on that copy of the database.

## D   Experimental Protocol

### D.1   Preparation and Synchronisation of Sensors

Calibration and synchronization of the devices are done using the Event Annotation functionality of the OpenSignals (R)evolution platform. Before starting, the subjects are instructed to take a comfortable sitting position.

First, the wearable sensors are prepared. The BioSignalsPlux[*] acquisition system is mounted with the ECG sensor, the EDA sensor, and the piezoelectric respiration belt. The experimenter starts by placing 3 Ag/AgCl electrodes on the ribcage of the subjects to capture the ECG signal. The BioSignalsPlux ECG sensor is designed to record single lead ECG signals using 3 derivation configurations. Then, 2 Ag/AgCl electrodes are attached to the palm of the non-dominant hand of the subject to acquire the EDA signal. Finally, the experimenter helps the subjects put on the respiration chest-belt, and adjust it to their morphology – making sure the participants are as comfortable as possible wearing the sensors.

After setting up the electrodes, the device is connected to the OpenSignals (R)evolution platform for recording and streaming the physiological data, thus allowing the experimenter to observe a real-time reading of the signals. The wearable sensors start recording during this procedure, but no video or audio is recorded since the camera is not set up yet. The wearables are installed first to enable good electrodes/skin interfacing, as the gel of the Ag/AgCl electrodes can take some minutes to correctly hydrate the skin. To ensure accurate and low-noise data, the experimenter checks the sensors' wires placement, as well as the posture and position of the subject before the start of the experiment. He adjusts and fixes the wires of the sensors using medical tape so that the presence of motion artifacts in the data during the collection is minimized.

Next, the Logitech QuickCam Pro 9000 RGB with integrated microphone is prepared. The camera is adjusted such that each subject is recorded in the middle of the frame with a neutral background. The participants sit approximately 50cm from the microphone. The start of the video/audio recording is marked on the OpenSignals (R)evolution platform using the event annotation plug-in.

Finally, once all devices are set up and the participants are installed, the experiment starts. The experiment instructions are displayed on a screen placed in front of the participants. The whole experiment is timed, i.e. each task and instruction are shown for a predetermined time, that is identical for all subjects. The beginning of the experiment is indicated by a beep sound. Another event annotation is added at the beep. This ensures the synchronization of the video, audio, and physiological signals for each task of the experiment.

---

[*]biosignalsplux, PLUX wireless biosignals S.A. (Lisbon, Portugal)

## D.2 Task Order and Instructions

The experimental protocol of `StressID` was designed to have a total duration of 35 minutes while spanning a wide variety of tasks. It includes a 3-minute guided breathing task used as a non-verbal baseline, followed by a neutral verbal baseline, 2 emotional video clips, 7 interactive 1-minute tasks, and a 2-minute and 30-second relaxation task. The choice and design of stressors is based on several considerations described hereafter:

All stressors have been selected to elicit 3 different categories of responses; 1) stimulate the audiovisual cortex of the participants, 2) increase the cognitive load by soliciting attention, comprehension, mental arithmetic or multi-tasking abilities, and 3) elicit psycho-social stress leveraging on public speaking as a stressor.

Overall, the tasks of the experiment are short – which allows the participants to perform several tasks in a row without tiring or losing acuity by the end of the experiment.

All interactive tasks are designed to leverage time restriction as a stressor by having a strict requirement for a response in 1 minute – thus, after receiving instructions on the screen, the subjects see a ticking 1-minute clock during the execution of each task.

The order of the stressors is designed to be unexpected to the participants. Therefore the experiment alternates between subgroups of tasks (e.g. Counting3 does not come after Counting2).

The level of detail provided in the instructions as well as the duration of the instruction was also carefully thought to maximize levels of stress in the experiment, by preventing participants from preparing for the coming task. The exact text of instructions received by the subjects for each task is given below:

- **Breathing:** "Now breathe deeply and relax."
- **Baseline:** "Start counting *forward* from 1 for 1 minute out loud".
- **Video1:** "Watch the video"
- **Video2:** "Watch the video"
- **Counting1:** "Start counting *backward* from 100 subtracting 3 for 1 minute (as fast as you can)"
- **Counting2:** "Start counting *backward* from 1011 subtracting 7 for 1 minute (as fast as you can)"
- **Stroop:** "Say out lout as many font colors as you can in one minute"
- **Speaking:** "Explain what are your strengths and weaknesses in 1 minute"
- **Math:** "Answer to the following mental arithmetic questions in 1 minute"
- **Reading:** "Read the following text in 1 minute (you can read silently)", followed by; "Explain the text in details to us in 1 minute"
- **Counting3:** "Start counting *backward* from 1152 subtracting 3 for 1 minute while touching your thumb with your other fingers"
- **Relax:** "Watch the relaxing video"

Each of the 11 tasks is followed by self-assessment questions. The counting forward baseline section is not defined as a task, but is designed to keep the participants in a neutral affective state, therefore it is not coupled with any self-assessment. Additionally, participants answer a survey question at the end of the experiment and indicate the task they considered most stressful.

## D.3 Self-assessments

Each task is annotated using answers to 4 self-assessment questions. The first 2 questions establish the participant's perceived stress and relaxation levels on a 0-10 scale. The following 2 questions are based on the SAM [3], and establish the participants' valence and arousal on a 0-10 scale. Fig. 3 shows the self-assessment questions as presented to the participants during the experiment.

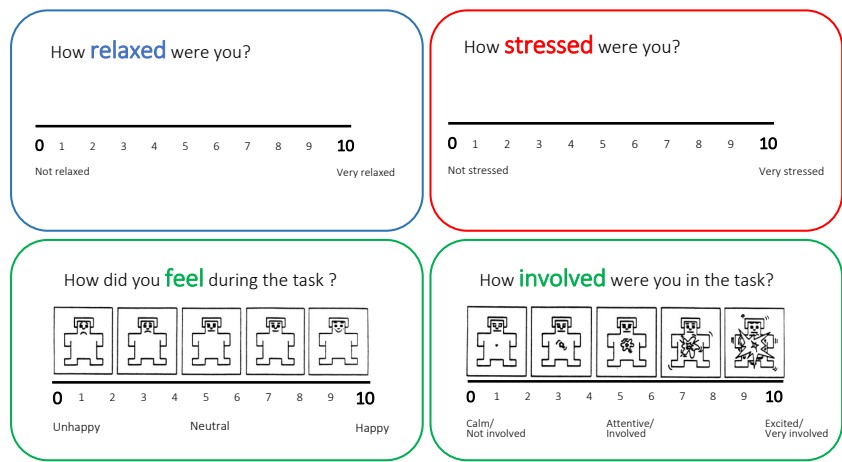

Figure 3: Illustration of the four self-assessment questions used in `StressID`.

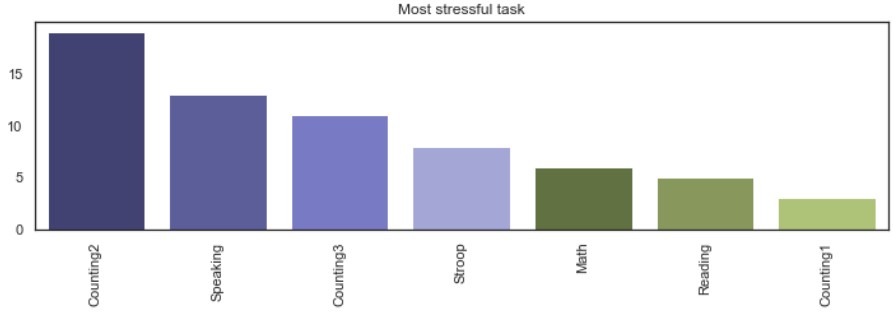

Figure 4: Most stressful tasks as designated by participants of `StressID`.

## E   Annotation Contents Analysis

### E.1   Survey: Most Stressful Task

We analyze the contents of the annotations of `StressID`. Fig. 4 shows the distribution of the answers to the question survey at the end of the experiment i.e. which task was perceived as most stressful for each subject. Approximately 30% of the participants of `StressID` designated the task **Counting2** as most stressful, 20% designated the task of public **Speaking**, 15% designated the task **Counting3**, while the remainder 35% chose between **Stroop**, **Math**, **Reading**, and **Counting1**. Although a majority of participants agreed on **Counting2** as the strongest stressor, this analysis highlights the advantages of relying on multiple and diverse stressors in an experimental protocol designed for stress identification. Perception of stress and relaxation can vary a lot from one participant to another – and more so, the effectiveness of a stressor can vary from one subject to another; while an arithmetic task can be a strong stressor for one individual, it can be an uneventful task for another.

### E.2   Participant-specific Distributions of `StressID` Annotations

We analyze the distributions of the stress, relaxation, arousal, and valence self-assessments for each participant of `StressID`. To have a global vision of the dataset, for each self-assessment question we represent on a single figure all subject-specific Kernel Density Estimate (KDE) plots in Fig. 5. The KDE plot, analogous to a histogram, represents the distribution of self-assessment data – only using a continuous probability density curve.

Several observations can be drawn from Fig. 5. First, for all 4 self-assessment questions, the participant-specific distributions are rather heavy-tailed, with the exception of a few subjects. This suggests that each participant of `StressID` gave a broad range of self-assessed scores across the experiment, highlighting the ability of the `StressID` protocol to elicit varied responses. Second, the perceived stress and relaxation levels of the participants across the experiment are well balanced, suggesting the experimental protocol enabled the creation of a dataset with proportional instances of stress and relaxation. Finally, we observe that the distribution of arousal scores is significantly skewed towards higher ratings across the dataset, highlighting the protocol's ability to create a high involvement in the participants and elicit strong responses – whether stress or amusement.

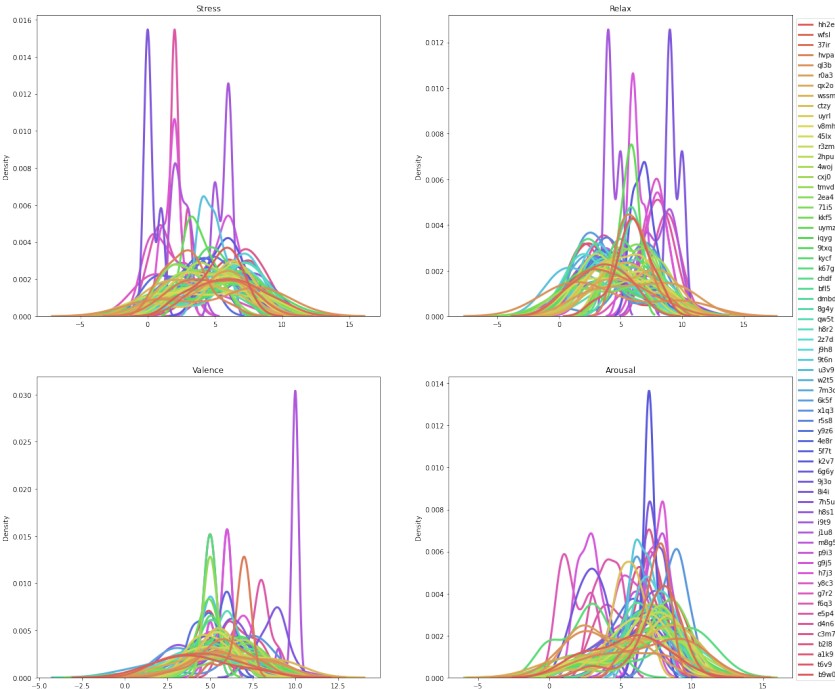

Figure 5: Participant-specific KDE plots for each of the self-assessment questions.

### E.3 Joint Distributions of `StressID` Annotations

We analyze the pair-wise joint distributions of the `StressID` annotations in Fig. 6. The analysis highlights a linear relation between stress and relaxation levels. In our experimental protocol, the participants' perceived levels of relaxation and stress associated with each task are mutually exclusive – globally, a subject cannot be both relaxed and stressed during a task. In addition, Fig. 6 highlights a positive correlation between stress and arousal, and a negative correlation between stress and valence – suggesting that across subjects and tasks, high arousal and low valence are associated with a higher level of stress. Similarly, relaxation is positively correlated to valence, and negatively correlated to arousal – suggesting low arousal and positive valence corresponds to higher levels of relaxation. These last observations are consistent with psychological studies [4, 10] describing stress on the circumplex model of affect [11], thus once again affirming the coherence of the `StressID` dataset.

## F   Stress and Emotion Identification

We train modality-specific pipelines to perform various classification tasks, e.g. discriminate between stressed and not stressed. In all the experiments, we generate 8 random splits, using 90% of the subjects for training, and 10% for testing for each split. The reported results are averaged over the 8 repetitions. The performances of the models are assessed using the f1-score on the test data.

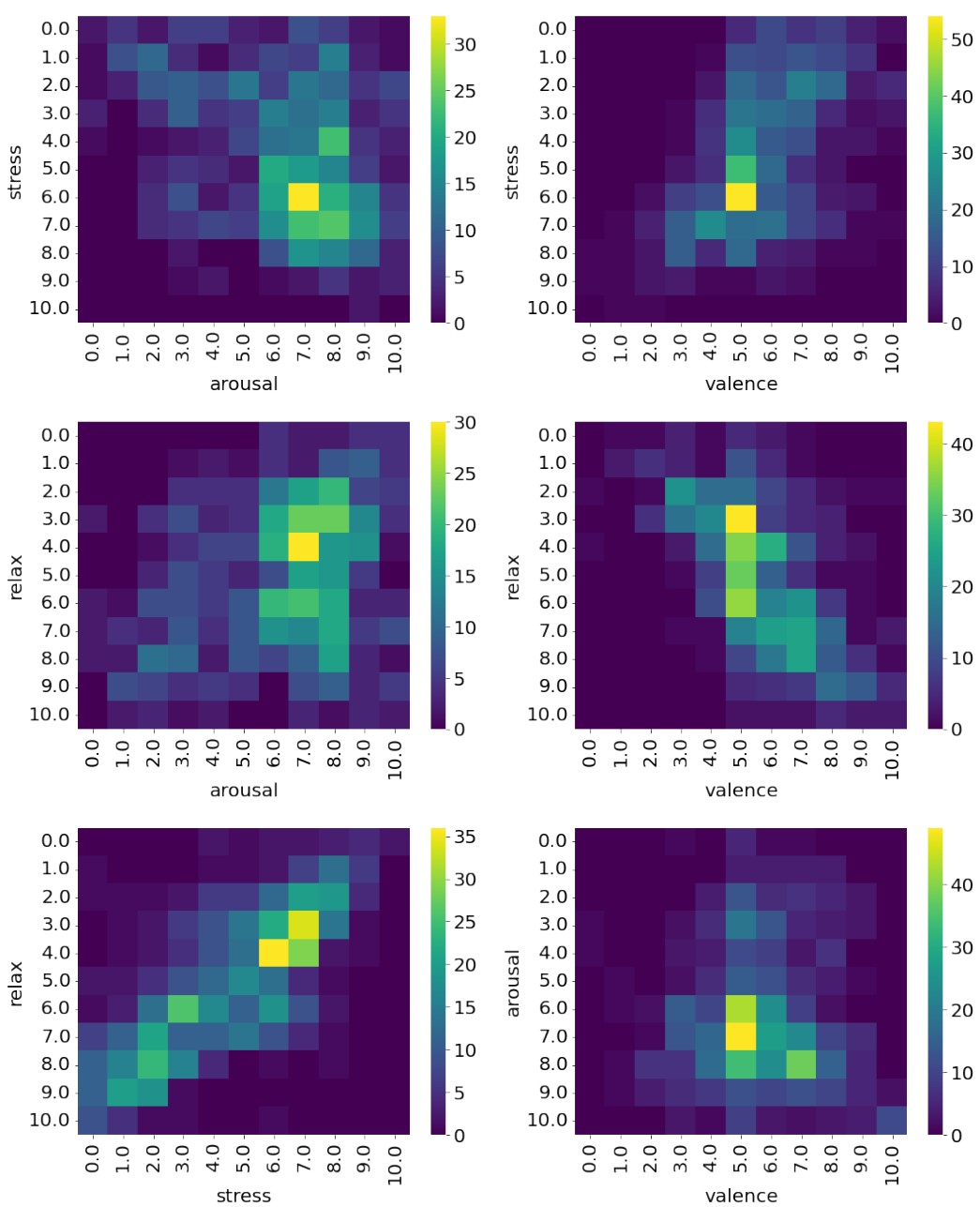

Figure 6: Joint distributions of pairs of self-assessment answers.

## F.1 Pre-Processing, Feature Extraction, and Classification

**Physiological Features.** In the first step, the ECG, EDA, and respiration signals are pre-processed to reduce high-frequency noise and baseline wander in the signals. Precisely, we use a 0.5 Hz high-pass Butterworth filter of order 5 for the ECG, a 5Hz low-pass Butterworth filter of order 4 for the EDA, and a 0.1-0.35 Hz bandpass Butterworth filter of order 2, followed by a constant detrending for the respiration signal. We use the `neurokit2` python package for all pre-processing. Then, 35 ECG features, 23 EDA, and 40 respiration features are extracted. These features include HRV measures, frequency features, and non-linear features. An exhaustive list of the features used in our baselines is provided in Table 2. Additionally, Fig. 7 illustrates an example of basic ECG, EDA, and respiration features visualized using `neurokit2`.

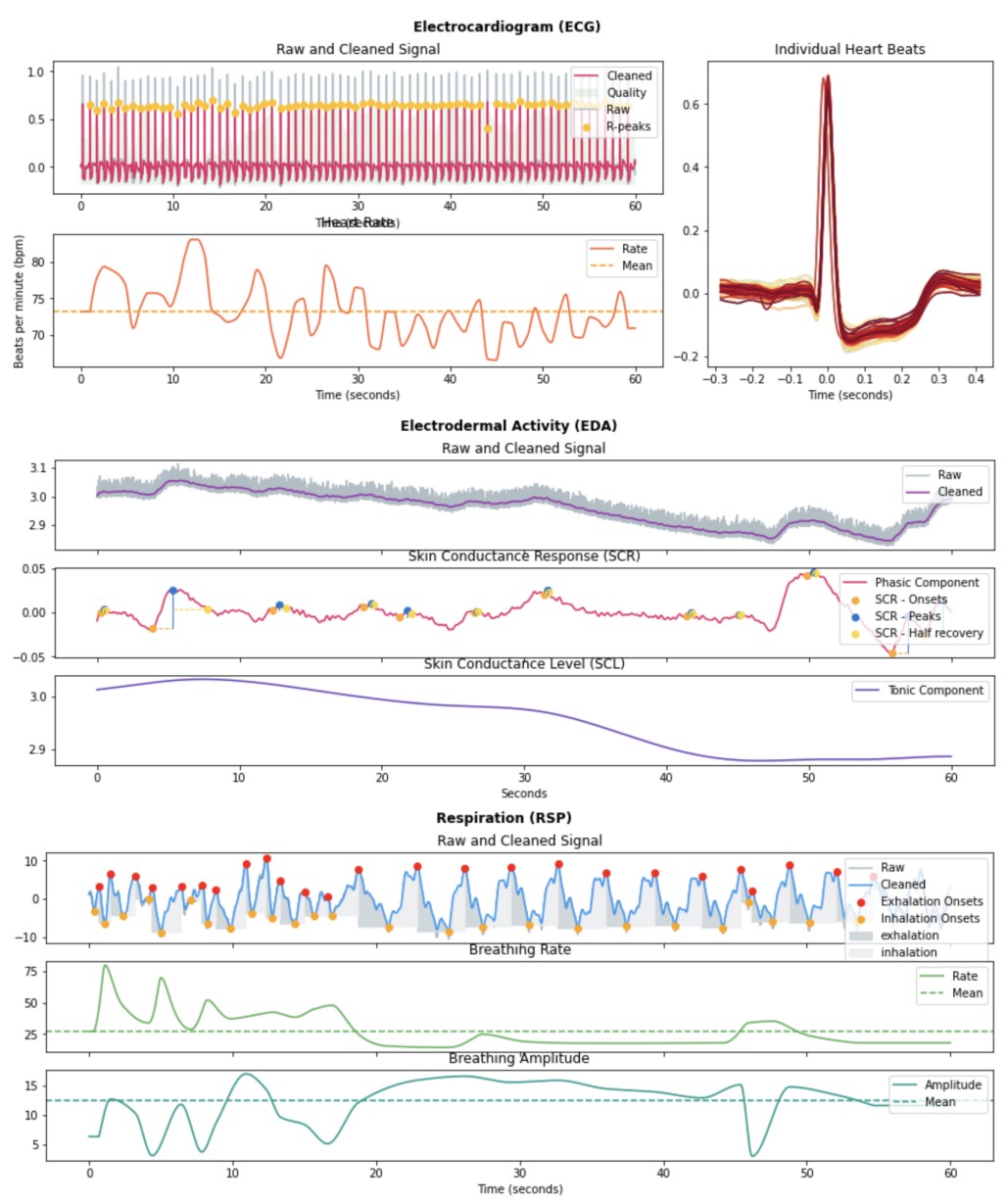

Figure 7: Example of features extracted from the ECG, EDA and respiration signals of `StressID`.

Table 2: Exhaustive list of physiological features extracted.

| Domain | Features | Total |
|---|---|---|
| **ECG** | | |
| **Time** | MeanHR, minHR, maxHR, stdHR, modeHR, nNN, meanNN, SDSD, CVNN, SDNN, pNN50, pNN20, RMSSD, medianNN, q20NN, q80NN, minNN, maxNN, triHRV | 19 |
| **Frequency** | Total power of the signal, LF, HF, LF/HF, ULF, VLF, VHF, rLF, rHF, peakLF, peakHF | 11 |
| **Non-linear** | SD1, SD2, SD1SD2, ApEn, SampEn | 5 |
| **EDA** | | |
| **Statistical** | MinEDA, maxEDA, meanEDA, std, skeweness, kurtosis, median, dynamical range, minSCR, maxSCR, meanSCR, stdSCR, minSCL, maxSCL, stdSCL, slopeSCL | 15 |
| **Time** | nSCRpeaks, area under SCR, mean amplitude SCR (meanAmp), maxAmp, mean response SCR (meanResp), sumAmp, sumResp | 8 |
| **Respiration** | | |
| **Time** | MeanRR, minRR, maxRR, stdRR, nBB (breath to breath), meanBB, SDSD, SVNN, SDNN, RMSSD, minBB, maxBB, meanTT (trhough to through), SDTT, minTT, maxTT, meanBA (breath amplitude), SDBA, minBA, maxBA, meanBW (breath width), SDBW, minBW, maxBW | 25 |
| **Frequency** | Total power, LF, HF, VLF, VHF, LF/HF, rLF, rHF, peakLF, peakHF | 10 |
| **Non-linear** | SD1, SD2, SD1SD2, ApEn, SampEn | 5 |

**Video Features.** We extract Action Units (AU) and eye gaze from each video frame using the OpenFace library [1]. The feature extraction from 587 videos is done in 3 hours 42 minutes using two Dual CPU Intel Xeon E5-2630 v4 processors. Fig. 8 is an example of AUs extracted on a subject of `StressID`.

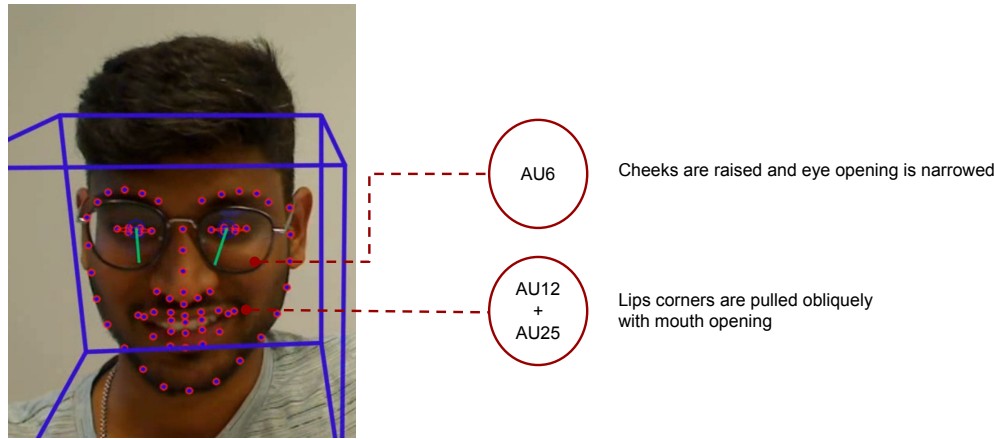

Figure 8: Example of AUs extracted from a video extract of `StressID`.

**Audio Features.** In the first step, amplitude-based voice activity detection (VAD) [8] is applied to the audio signals prior to feature extraction to eliminate non-speech segments. We first extract handcrafted (HC) features, such as MFCCs, using the `libROSA` python package [9]. Fig. 9 is an example of MFCCs extracted on a subject of `StressID`. Additionally, DNN-based feature extraction is performed using a large pre-trained Wav2Vec (W2V) model [12]. Features are extracted every 20 ms and are averaged over time to obtain a single 513-component embedding per utterance. The extraction is done using a GeForce RTX 3090 graphic card.

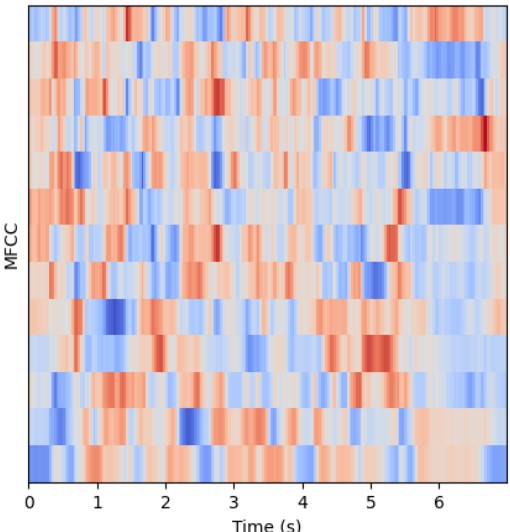

Figure 9: Example of MFCCs features extracted from an audio data of `StressID`.

**Classification.** For all baselines, we have evaluated several combinations of feature selection algorithms and classifiers and selected the best-performing ones for our baseline results.

For feature selection, we evaluated a Recursive Feature Elimination (RFE) algorithm, an L1 regularisation, and Principal-Component Analysis (PCA) for dimension reduction, as well as no feature selection. For the classification models, we have considered a large range of classical classifiers with different hyper-parameterizations. The exhaustive list is reported in Table. 3

### F.2 Additional Experiments

### F.2.1 Emotion Recognition.

We report here additional experiments performed with binary labels extracted from the 4 self-assessments. We evaluate our learning pipeline on 4 binary classification tasks; namely discriminate between stressed (1) vs not stressed (0), relaxed (1) vs not relaxed (0), high valence (1) vs low valence (0), and high arousal (1) vs low arousal (0).

**Labels.** Each continuous value of the self-assessment is split as follows; if *value* is less than 5 then the label is 0, and if *value* is equal or greater than 5, then the label is 1. The created **stress** label is balanced and composed of 48% and 52% of class 0 and 1 respectively. Similarly, the **relax** label is composed of 54% and 46% of 0 and 1 respectively, and the **valence** label consists of 50% of each class. On the other hand, the **arousal** label is severely imbalanced and consists of 71% of high arousal (1) and 29% of low arousal (0).

**Results and Discussion.** The classification performances for all modalities and each label are reported in Table 4. Our analysis confirms that the labels and the acquired data are coherent and meaningful, and the labels are predicted from the data with f1-scores well above the random.

Despite the different number of trials for each modality, some general observations can be highlighted. The valence appears here as the most difficult label to predict. This is especially true for audio and video, while physiological data seems to carry more useful information to discriminate between positive and negative valence. For the video, this can be related to the fact that a positive or negative valence in this set-up can be expressed with similar expressions. A person can smile because they are amused by the task or they can smile nervously. Recognizing a positive smile from a negative one is still a challenging task to this day in the field of emotion recognition.

On the other hand, the arousal is better predicted by the audio. This can be due to the fact that when people are more engaged in the task their tone of voice is incremented.

Table 3: List of tested classifiers and corresponding grid search of hyper-parameters.

| Model | Hyper-parameters | Grid search values |
|---|---|---|
| **Support Vector Machines** | kernel | 'linear', 'rbf', 'sigmoid' |
| | C | 0.1, 1.0, 10.0 |
| | gamma | 'scale', 'auto' |
| **K-Nearest Neighbors** | n_neighbors | 3, 5, 10, 20 |
| | weights | 'uniform', 'distance' |
| | algorithm | 'auto', 'ball_tree','kd_tree', 'brute' |
| **Random Forests** | n_estimators | 100, 150, 200 |
| | criterion | 'gini','entropy' |
| | max_depth | 3, 5, 7 |
| | min_samples_split | 2, 4, 6 |
| | min_samples_leaf | 1, 2, 3 |
| | max_features | 'auto', 'sqrt', 'log2' |
| | class_weight | None, 'balanced', 'balanced_subsample' |
| **Multi Layer Perceptron** | activation | 'logistic', 'tanh', 'relu' |
| trained using a cross-entropy loss | alpha | 0.0001, 0.001, 0.01 |
| in combination with an Adam [7] | solver | 'lbfgs', 'adam' |
| optimizer and number of hidden | learning_rate | 'constant', 'invscaling', 'adaptive |
| layers in [2,3,4], layer width | shuffle | True, False |
| in [64, 128, 256] | momentum | 0.7, 0.8, 0.9 |
| | early_stopping | True, False |
| **Logistic Regression** | penalty | l1, l2 |
| | C | 0.1, 1, 10 |
| | solver | 'liblinear', 'saga |
| **Gradient Boosting Classifier** | loss | 'deviance', 'exponential |
| | n_estimators | 100, 150, 200 |
| | learning_rate | 0.1, 0.5, 1.0 |
| | max_depth | 3, 5, 7 |
| | min_samples_split | 2, 4, 6 |
| | max_features | 'sqrt', 'log2 |
| **LGBM** | boosting_type | 'gbdt', 'dart |
| | importance_type | 'split', 'gain' |
| | num_leaves | 20, 30, 40 |
| | max_depth | 5, 10, -1 |
| | learning_rate | 0.1, 0.01 |
| | n_estimators | 100, 200 |
| | objective | 'binary', 'multiclass' |
| | metric | 'binary_logloss', 'multi_logloss' |
| | colsample_bytree | 0.8, 1.0 |
| | reg_lambda | 0.5, 1. |
| | reg_alpha | 0.0, 0.5 |
| **Ridge Classifier** | alpha | 0.1, 1.0, 10.0 |
| **Decision Tree Classifier** | criterion | 'gini', 'entropy' |
| | max_depth | None, 3, 5, 7 |
| | min_samples_split | 2, 4, 6 |
| | min_samples_leaf | 1, 2, 3 |
| | max_features | 'auto', 'sqrt', 'log2' |

For the tasks of identifying stress and relaxation, the physiological signals appear as the most meaningful modality. Nonetheless, the results highlight good performances for all modalities, highlighting the strong correlations between the recorded data and the labels.

## F.2.2  Multimodal Learning on Other Multimodal Combinations

To further highlight the advantages of multimodal learning, we have evaluated multimodal baselines on all the possible modality combinations of the available data, i.e. physiological/video only, video/audio only, and physiological/audio only. Each baseline that combines the features extracted from different modalities is evaluated on all the data available in the subset of tasks featuring the said modalities. When the subset presents a strong imbalance in the labels, we use Minority Over-sampling

Table 4: Baseline f1-scores for different classification tasks. Each unimodal baseline is trained and tested on all available tasks of the corresponding modality (#tasks).

| Data subset (#tasks) | Binary stress | Binary relax | Binary arousal | Binary valence |
|---|---|---|---|---|
| Physiological (711) | $0.73 \pm 0.04$ | $0.67 \pm 0.06$ | $0.66 \pm 0.06$ | $0.64 \pm 0.07$ |
| Video (587) | $0.62 \pm 0.04$ | $0.62 \pm 0.06$ | $0.67 \pm 0.10$ | $0.54 \pm 0.07$ |
| Audio-HC (385) | $0.67 \pm 0.04$ | $0.62 \pm 0.1$ | $0.79 \pm 0.09$ | $0.55 \pm 0.09$ |

Table 5: Performances of multimodal baselines for the classification of stress, compared to unimodal models.

| Modalities (subset size) | 2-class | | 3-class | |
|---|---|---|---|---|
| | F1-score | Accuracy | F1-score | Accuracy |
| Physiological (711) | $0.73 \pm 0.02$ | $0.72 \pm 0.03$ | $0.55 \pm 0.04$ | $0.56 \pm 0.03$ |
| Video (587) | $0.70 \pm 0.03$ | $0.70 \pm 0.03$ | $0.55 \pm 0.03$ | $0.55 \pm 0.03$ |
| Audio (385) | $0.70 \pm 0.02$ | $0.66 \pm 0.03$ | $0.56 \pm 0.04$ | $0.52 \pm 0.04$ |
| Physiological + Video (587) | $0.72 \pm 0.04$ | $\mathbf{0.72 \pm 0.04}$ | $0.62 \pm 0.05$ | $0.52 \pm 0.07$ |
| Video + Audio (370) | $\mathbf{0.76 \pm 0.05}$ | $0.68 \pm 0.05$ | $0.52 \pm 0.06$ | $0.45 \pm 0.05$ |
| Physiological + Audio (385) | $0.68 \pm 0.08$ | $0.62 \pm 0.07$ | $0.50 \pm 0.07$ | $0.41 \pm 0.07$ |
| All modalities (370) | $0.72 \pm 0.05$ | $0.65 \pm 0.05$ | $\mathbf{0.63 \pm 0.05}$ | $\mathbf{0.58 \pm 0.07}$ |

Techniques (SMOTE) to balance the training set in each of the 10 repetitions, and leave the test sets untouched.

The multimodal baselines are compared with the best-performing unimodal baselines, trained on all the available data of each modality. The results for all multimodal baselines for the 2-class and 3-class classification are reported in Table 5. As observed in the results of Section 4, multimodal models using decision-level fusion show considerable improvement over the performances of unimodal models. We, therefore, evaluate models based on SVMs merged with different decision rules (i.e. sum, product, average, or maximum rule) and report the best-performing ones here.

Despite the different subset sizes for each baseline, some conclusions can be drawn. First, the performances of unimodal baselines highlight that the physiological modality carries more information for the classification of 2-class stress. However, all 3 unimodal baselines achieve comparable results for the classification of 3-class stress. It can be noted that the baseline on the physiological modality shows slightly better performances in terms of accuracy, suggesting physiological data is more susceptible to carrying information allowing to discriminate between different emotional states.

Second, the multimodal baselines show that combining multiple modalities by merging the results of unimodal models using late voting on each modality (decision fusion), considerably improves classification performances. For 2-class prediction, the best performance is shown by the combination of video and audio features. However, for 3-class classification the best performance is achieved when combining all available modalities, highlighting the predictive potential of combining multiple complementary sources of data, and showing once more the importance of physiological data in the discrimination between a relaxed state and acute stress.

### F.2.3 Investigating the Effect of Gender Imbalance

A balanced dataset is crucial for performing bias-free analyses and minimizing the risk of bias in algorithm development. Potential imbalances in gender, race, age, or background of the participants can limit the development of fair and equitable applications, and researchers need to be aware of this aspect. To sensitize users to this issue, we have evaluated the predictive potential of our dataset on a subset of StressID presenting a balanced ratio of female and male subjects – using the previously introduced unimodal and multimodal baseline models. To create this subset, the recordings from the 18 female participants are kept untouched, and only 18 male participants are randomly selected, thus resulting in a subset composed of 36 subjects. The classification performances for unimodal and multimodal baselines for the 2-class classification are reported in Table 6. The results are averaged over 5 random balanced subsets built this way. All baselines are performed on tasks available for all

Table 6: Performances of unimodal and multimodal baselines for the 2-class classification of stress, using a gender balanced subset. The high variability in the results is explained by the use of different random subsets of the data across repetitions.

| Modalities | F1-score | Accuracy |
|---|---|---|
| Physiological | $0.69 \pm 0.1$ | $0.62 \pm 0.1$ |
| Video | $0.73 \pm 0.07$ | $0.65 \pm 0.08$ |
| Audio | $0.69 \pm 0.07$ | $0.64 \pm 0.07$ |
| All modalities (370) | $0.73 \pm 0.07$ | $0.64 \pm 0.07$ |

3 modalities. When the subset presents an imbalance in the labels, we use SMOTE to balance the training set, and leave the test sets untouched.

Two important conclusions can be drawn from the results observed in Table 6. First, several baselines built on balanced subsets outperform (or compare to) the baselines using all available data. This suggests more balanced datasets can improve the performances of subsequent models, and thus highlights potential bias induced by imbalanced representation in data. This is to be expected, as training on well-balanced data decreases the risk for a model to overfit – which in the case of gender imbalance can be translated as learning on male subjects mainly during the training phase and performing poorly during the testing phase on female subjects, less seen during training. With this experiment, we aim to increase the awareness of users to the effect of gender imbalance in particular, and we invite them to account for this possibility in their analyses.

Second, these results illustrate the possibility of developing algorithms achieving good classification performances on restricted parts of `StressID`. Indeed, our dataset offers the possibility to focus on particular subsets of the data while still ensuring good prediction scores, thanks to its large total population. For instance, the gender-balanced subset of our dataset represents 36 participants, while similar multimodal datasets collect data from less than 30 participants in total, without eliminating the limitation of gender imbalance – thus highlighting once more the advantages of `StressID`.

We strongly encourage users of `StressID` to anticipate possible consequences by taking the appropriate steps to build equitable systems before their use in real-life applications – this experiment provides a good starting point and example of how to proceed.