# OpenReview forum: "StressID: a Multimodal Dataset for Stress Identification"
_NeurIPS.cc/2023/Track/Datasets_and_Benchmarks — NeurIPS 2023 Datasets and Benchmarks Poster_

### Official Review · Reviewer_qPpi · 2023-06-29
**A new multimodal stress dataset**

**Rating:** 7
**Confidence:** 3

**Strengths:**

1. The dataset is not specifically designed for a particular scenario, and it has strong versatility.

2. The dataset includes video and audio data, synchronously collected physiological signals such as ECG, EDA, RR. In addition, derived indicators based on physiological signals were calculated, such as HRV. These indicators play an important role in distinguishing stress levels; therefore the modalities of the dataset are comprehensive.

3. The annotation of the dataset uses the SAM method with participants self-reporting by answering four questions. The authors then provide binary and ternary labels based on participants' responses which expand the application scenarios of this dataset.

4. The paper presents some baseline models for multimodal feature fusion that serve as good examples for algorithm development.

**Additional Feedback:**

For the videos in the dataset, I would like to know why the frame rate is 5? This is a very low frame rate, which may reduce the value of video modality. As far as I know, the camera used in the paper supports 30fps recording.

**Clarity:**

The article is generally clear, but there are some minor issues, and the language needs to be polished. Section 3.1.1 describes the experimental procedure, where the first and fourth blocks directly describe the task (although these two blocks only contain one task), while other blocks introduce the block before introducing tasks, which may be slightly confusing in terms of format.

There is an error in line 224's description of HRV; pNN50 should correspond to intervals differing by 50ms instead of 50s.

Additionally, there are some sentence issues; for example, "It is contains" in line two should be changed to "It contains," and this is not the only mistake present.

**Correctness:**

This dataset uses the BioSignalPlux system to collect physiological signals, uses a network camera to collect video and audio signals, and the data annotation is done using the self-report method of the subjects, which are all common methods in previous work. The paper is basically correct in terms of dataset collection, but I have some doubts about the baseline algorithm of this article. It may not have correctly integrated multimodal signals.

**Documentation:**

The paper mentioned the hosting plan for the dataset, and supplementary materials include a summary and purpose of the dataset, as well as a review mechanism to ensure responsible data usage. Unfortunately, I cannot access the dataset using the provided account and password from the author; my browser shows a 500 error. I am not sure if it is an issue with server-side or browser settings.

**Ethics:**

I believe that there are no additional ethical concerns in this article, the collection and publication of the dataset have been approved by the ethics committee, and the authors have anonymized the data to protect the privacy of participants.

**Limitations:**

This article describes the Limitations, which is good.

The dataset was collected in a carefully arranged laboratory environment, with the stress levels of the subjects controlled by induction materials provided by the experimental protocol, and data annotation determined by subject self-reporting on scales. Therefore, this dataset mainly annotates short-term stress, while stress levels may also be influenced by both short-term and long-term stress; this dataset cannot assess long-term stress levels. The induction materials may have different effects on each individual, and the subjects' reports of their stress levels are subjective; this subjectivity may affect annotation accuracy.

**Opportunities For Improvement:**

The main weakness is that the provided multimodal fusion baseline model does not reflect the advantages of this dataset. Ideally, the classification accuracy of multimodal fusion should be better than any single modality. However, according to the results in Table 2, Physiological modality is the best, followed by Audio modality, and multimodal accuracy is not good. This may be due to the algorithm used not being carefully designed and having poor effects on modality fusion; however, it cannot be ruled out that it was caused by a lack of strict synchronization during data collection in the dataset itself, which may make multimodal fusion difficult. Therefore, improvements should be made to algorithms to demonstrate effectiveness in terms of multi-modal fusion for this dataset as much as possible.

**Relation To Prior Work:**

The connection with past work is close, and the paper provides more comprehensive multimodal data, including video, audio, and synchronous recording of three physiological signals, which helps to study multimodal fusion algorithms.

**Summary And Contributions:**

The author proposed a multimodal stress identification dataset, including video, audio, electrocardiogram (ECG), electrodermal activity (EDA), and respiration rate (RR). The author induced different stress levels in the participants through a designed experimental procedure and reported the stress levels through self-assessment. The author provides some easily reproducible algorithms as baselines for multimodal fusion, which are derived from previous work but have been modified to integrate features from multiple modalities.

---

> ### Author Response · Authors · 2023-08-21
>
> Dear reviewer,
>
> We would like to thank you for your very valuable comments, and let you know we have updated our paper to address them. We summarize our revisions below:
>
> #### **Opportunities For Improvement:**
> - **Multimodal baselines.** In this new version, we have added baselines, and improved the multimodal ones. We have implemented models employing decision-level fusion following [1],[2]. We have reported additional experiments in section 4.2 allowing us to directly compare the multimodal baseline to the unimodal ones (c.f. Table 3), by training all models on the same subset of 370 tasks featuring all modalities. These new results reported in the global answer to reviewers, highlight the advantages of multimodal learning for stress recognition more clearly.
> - **Synchronization.** While the synchronization of the modalities in our dataset can be prone to human errors (of a few ms), the recorded data has been carefully and manually checked to ensure a high quality dataset.
>
> #### **Clarity:**
> - We have revised the manuscript thoroughly to make it more clear. Among these and as suggested, we have modified the formatting of Section 3.1.1., hoping to make the description of our experimental protocol more clear and easy to follow.
> #### **Documentation:**
> - We are sorry to hear you’ve encountered an error when downloading the dataset. An issue was reported earlier during the review process. We addressed it with the help of our technical office, after which the AC confirmed it as sorted (see official comments thread below). If the advice provided does not help, we would be happy to work with you to resolve your issue.
>
>
> We hope we have better highlighted the contributions our dataset can bring to the advancement of stress identification. We will be happy to answer any additional questions.
>
> **Additional references:**
>
> [1] Fen Xu and Zhe Wang. Emotion recognition research based on integration of facial expression and voice. 2018.
>
> [2] K Prasada Rao, MVP Chandra Sekhara Rao, and N Hemanth Chowdary. An integrated approach to emotion recognition and gender classification. 2019.

---

> > ### Comment · Reviewer_qPpi · 2023-08-28
> > **My main concerns have been addressed.**
> >
> > My main concerns have been addressed, but the author does not seem to answer why the frame rate of this dataset is only 5? Especially when the camera used for collection supports 30fps.
> >
> > I changed the rating from 6 to 7 to encourage the author's work.

---

> > > ### Author Response · Authors · 2023-08-29
> > >
> > > We thank you for your support and apologize for missing this question initially.
> > >
> > > The videos were indeed recorded at 15fps (the maximal frame rate allowed by the Logitech software on the computer used for recording). As a pre-processing step in our baseline experiments, we have however downsampled the recordings to 5 fps prior to Action Units extraction, thus finding a compromise between classification performance and execution time. We thank you for pointing this issue out, and we have now corrected this information in our paper.

---

> > > > ### Comment · Reviewer_qPpi · 2023-08-29
> > > >
> > > > Thank you. To my knowledge, if the transmission format is changed to MJPG, it can record at 30fps. Providing a high frame rate version would be helpful.

---

### Official Review · Reviewer_jqtB · 2023-07-07
**Queries to answer/Lots of further work needed**

**Rating:** 5
**Confidence:** 3
**Clarity:** The paper lacks in clarity.

**Strengths:**

The given dataset captures multimodal data related to stress, including audio, video, and physiological signals. These different types of data provide complementary information that can be used to build a more complete understanding of stress.

**Additional Feedback:**

Authors need to discuss more on the comparison with the state of the art.

**Correctness:**

The claims made are questionable as the baseline model provided in support of the dataset is not sufficient to show the applicability of the dataset. Also the model accuracies are not reported.

**Documentation:**

Yes

**Limitations:**

1)The use of a multimodal approach to stress detection, which involves attaching a number of electrodes to the subject, could potentially cause stress in itself. This is because the sensors may be uncomfortable or restrictive, and the subject may feel self-conscious or anxious about being monitored. Additionally, the act of attaching the sensors may itself be stressful, as it may remind the subject of a medical procedure or other stressful event. This dataset cannot be related to real world scenarios.

2)When I tried to download the dataset...I am being asked lot of questions...Is the dataset available for download

3)Why the need to record so many signals..seems like overkill

4)Table 2 is not clear…why are tasks number varying physiolo(715), video(605) and so on

5)Why is multimodal classification low..Multimodal classification means all recording  signals are used for classification right?




**Opportunities For Improvement:**

•	Comparison shown with the existing datasets are not sufficient, more needs to be done.
•	More details of the provided multimodal baseline model is needed.
•	Why only MLP was trained and no other model?


**Relation To Prior Work:**

The paper lacks sufficient discussion with prior works and existing stress datasets

**Summary And Contributions:**


The authors presented StressID, a new dataset that can be used to identify stress. The dataset includes video, audio, and physiological data, which are collected using a variety of sensors. The data is high quality and low noise, and it is synchronized across all modalities. StressID also includes baseline models for stress classification, as well as a cleaning, feature extraction, feature selection, and classification pipeline. This pipeline has been suggested to train a multimodal predictive model for stress identification. Additionally, an original multimodal dataset was included, which focused on tasks designed to induce stress. It incorporated ECG, EDA, respiration, facial video, and audio recordings, all synchronized and annotated with self-assessments from participants, evaluating their levels of relaxation, stress, valence, and arousal.

---

> ### Author Response · Authors · 2023-08-21
>
> Dear reviewer,
>
> Thank you very much for your feedback. We address your comments below :
>
> #### **Opportunities For Improvement:**
> - **Comparison to state of the art.** In our revised manuscript, we have updated the “Comparison with state of the art” section in the related work (Section 2) to better highlight the advantages of our dataset over other similar ones. $\texttt{StressID}$ aims to address the common limitations found in similar datasets, and fill the gaps in the existing related work.
> - **Classification models.** We would like to clarify that in our first version, we have reported the results of the best performing models for each baseline, although several classification models have been trained – more information on the experiments is available in Appendix F.1. However, we understand it can be relevant to compare the performances of multiple classifiers. We have therefore revised section 4, and added several additional models commonly used in the state of the art in the field (Table 2). Additionally, we have added section 4.2 explaining better the multimodal approach.
>
> #### **Limitations:**
> - **(1) Limitations.** We are aware $\texttt{StressID}$ is not necessarily representative of real life situations, for several reasons. Indeed, we report this as a limitation of our dataset (Section 5.). Regarding electrode placement, we thank you for bringing to our attention the stressful aspect it may induce on the subject itself. We have added this as an additional limitation. Nonetheless, by starting the experiment with a guided breathing exercise intended for resetting their affective states to neutral, we minimize, as much as possible, variability due to external factors susceptible to impact the mental state of participants.
> - **(2) Data access.** We are sorry to hear you’ve had issues with the access to our dataset. We have designed the request form on our website, and the credentialization process to be compliant to European law and GDPR. This information is required to be compliant with the law and thus it should not be seen as a limitation nor it should penalize our work. If however, you’ve had issues downloading the dataset with the sets of credentials provided to the reviewers, we invite you to refer to the official comments thread below, including help from our technical office concerning a download issue that has been reported and solved, as confirmed by the AC. If the advice provided does not help, we would be happy to work with you to resolve your issue.
> - **(3) Multiple modalities.**  Multimodal learning has been shown to have considerable advantages [1]. The aim of multimodal stress recognition is to use the combined knowledge extracted from each modality to build a stronger and more robust system [2]. Diverse modalities carry complementary information: video and audio capture the behavioral component of emotions, the reactions that are visible from outside, while the physiological signals capture valuable internal states not visible on camera such as cardiac activity, or skin sweating. By providing access to multiple synchronized modalities, $\texttt{StressID}$ enables cross-modal analyses and has the potential of improving the understanding of the relationships between video, audio, and physiological responses to stress. We discuss these aspects in the conclusion to our work (Section 7).
> - **(4) Subset sizes.** Indeed, several video and audio recordings have been damaged during the collection process, thus only the physiological modality is available for these subjects. Moreover,  multiple tasks in $\texttt{StressID}$ are non-verbal so the audio component is naturally composed of fewer recordings. The unimodal baselines are all trained on all the data  available for each modality – which explains the varying sizes of subsets in Table 2. Please refer to Section 3.2.2 where we discuss these elements of the dataset composition.
> - **(5) Multimodal baseline performances.** In our initial results, the multimodal model performed worse than unimodal ones because it was trained on a significantly smaller subset. In this version, we have added additional experiments (section 4.2) to directly compare the multimodal baseline to the unimodal ones (c.f. Table 3), by training all models on the same subset of 370 tasks that have no missing modalities. These new results are reported above in the global answer to reviewers, and better highlight the advantages of multimodal learning for stress recognition more clearly.
>
> We hope we were able to clarify some points for you, and we would be happy to answer any other question you might have regarding the paper or dataset access.
>
> **Additional references:**
>
> [1] Yu Huang, Chenzhuang Du, Zihui Xue, Xuanyao Chen, Hang Zhao, and Longbo Huang. What makes multi-modal learning better than single (provably). 2021.
>
> [2] Naveed Ahmed, Zaher Al Aghbari, and Shini Girija. A systematic survey on multimodal emotion recognition using learning algorithms. 2023.

---

> > ### Comment · Reviewer_jqtB · 2023-08-29
> > **Thanks for replies**
> >
> > The replies seems reasonable however I would not change my rating..Please program chairs consider  all reviewer comments and decide about acceptance...Best wishes

---

### Official Review · Reviewer_QzsT · 2023-07-21
**A high-quality dataset, yet with limited contributions.**

**Rating:** 5
**Confidence:** 4

**Strengths:**

- The data collection process is well-designed in general. I appreciate the authors' effort in collecting such a dataset.
- The paper has a very clear structure. The writing has high quality and is very easy to follow.

**Additional Feedback:**

Please see my comments above.

**Clarity:**

The paper is written well and very easy to follow. The figures clearly convey the message.

**Correctness:**

Assuming my questions about study details can be addressed, the dataset is constructed with high quality. However, my main concern is about the contribution of the dataset.

**Documentation:**

The dataset includes clear documentation.
The code has a clear structure and supports easy reproducibility.

**Ethics:**

The authors has discussed it appropriately.

**Limitations:**

The authors openly discussed the limitation of their work.

**Opportunities For Improvement:**

- The contribution of the dataset is limited, given the richness of previous work.
While I appreciate the authors' great efforts in collecting the dataset, the novelty of the dataset and the contribution it brings to the community is limited. As the authors summarize in Table 1, there have been a number of datasets available in the field. It's hard to identify the significant new aspects of StressID.
(a) The number of participants is close to datasets such as CLAS and Distracted Driving dataset.
(b) From the modality perspective, StressID and other datasets collected signals from diverse sensors with several overlap sources.
(c) StressID employs stressors that are the combination of subsets of other datasets.
(d) The annotation process of StressID is also close to previous work, such as WeSAD, CLAS, and MuSE.
Although StressID is collected with high quality, none of these four aspects show the significant novel contribution of StressID.

- The definition of classification tasks seems to be vague.
For the benchmark results, the authors defined one binary classification and one three-class classification task. However, the threshold seems to be relatively arbitrary. It would be great if the decisions could be supported by the literature.

- Missing Study Details.
There are a few missing details in the study. For example, the authors claimed in Lines 122 - 123: "The order of the stressors is designed to be unexpected to the participants". It's unclear to me whether this means that the order is randomized across participants or all participants went through the same order (which would introduce the bias of order effect). The same concerns are also applicable to 2 video clips for emotion response elicitation.

- Baseline methods are overly simple, and their results are not comparable.
The authors present several standard/traditional signal processing processes for different modalities of the data. The multimodel baseline is overly simple by fusing the feature and training a very simple MLP (thus the low performance of the model is expected). There is no contribution of the modeling techniques. Meanwhile, since the number of tasks is different across different rows in Table 2, the comparison across different rows is not accurate.

**Relation To Prior Work:**

The paper covers a good amount of related work. Table 1 is very helpful in summarizing the existing datasets in the field.

**Summary And Contributions:**

In this paper, the authors collected a new dataset called StressID, which consists of multimodal data sources, including facial expression video, audio, as well as physiological signal recordings (ECG, EDA, and respiration signals). The datasets include 65 users performing 11 pre-determined tasks in a controlled environment. Using this dataset, the authors implemented several simple baseline machine learning models and accessible the data and code.

---

> ### Author Response · Authors · 2023-08-21
>
> Dear reviewer,
>
> Thank you very much for your valuable comments and the interest you’ve shown in our work. We have taken into account your feedback and have revised our paper accordingly. We address each of your comments below :
>
> #### **Opportunities For Improvement:**
> - **Contribution of the dataset.** While there exists other multimodal datasets similar to ours, such as MuSE, SWELL-KW or the distracted driving dataset, we consider each presents at least one major limitation (e.g. small number of participants for MuSE and SWELL-KW, and limited usage for the the distracted driving dataset). We aim to address these limitations, filling the gaps in the existing previous work; indeed $\texttt{StressID}$ is the first multimodal dataset (including both physiological and behavioral modalities) for stress identification recorded on a large number of participants – that also features a wide range of stimuli ensuring versatility of deriving applications. We have revised the “Comparison with state of the art” paragraph in the related work to better highlight the advantages of our dataset over other similar ones.
> - **Labels definition.** The definition of the labels used in our baselines experiments is proposed to illustrate the predictive potential of our dataset. An interesting feature of the dataset is that we provide the original unprocessed self-reportings of the participants, allowing users of $\texttt{StressID}$ to design any label they consider better fitted for their application. Moreover, we would like to highlight that the 3-class label is designed following psychological evidence[1],[2], which is confirmed in our analyses of the self-assessments distributions.
> - **Study details.** The order is designed to be unexpected, in the sense that same type tasks do not necessarily follow each other (e.g. counting3 does not come after counting2). The order of the stressors is the same across all subjects. However, participants are explicitly asked not to discuss the contents of the experiment. The answers to the survey question about the most stressful task (reported in Appendix E.1), suggest the experiment is not subject to strong order or recency bias.
> - **Baselines and comparisons.** In our new version, we have included additional models in line with the state-of-the-art in the field, and improved the performance of multimodal models (Section 4). We have also added experiments to directly compare the multimodal baseline to the unimodal ones (c.f. Table 3), by training all models on the same subset. These new results are reported above in the global answer to reviewers, and we consider they reflect better the advantages of multimodal learning.  Nonetheless, we would like to highlight that the goal of this work is not about making novel model contributions. The presented models act as baselines that can be used by researchers as a reference starting point  in their research work..
>
> We hope we were able to highlight the value of our dataset, and the contributions it can bring to the advancement of stress identification. We will be happy to answer any additional questions.
>
> **Additional references:**
>
> [1] Sven-Åke Christianson. Emotional stress and eyewitness memory: a critical review. 1992
>
> [2] Maria D McManus, Jason T Siegel, and Jeanne Nakamura. The predictive power of low-arousal positive affect. 2019

---

> > ### Comment · Reviewer_QzsT · 2023-08-29
> > **Thanks for the response**
> >
> > I appreciate the authors' response and clarification of my questions. I agree that the update makes the paper stronger. However, my concern about the limited novelty of the work is still there. I do acknowledge the uniqueness of the dataset, yet its contribution is relatively marginal. I will keep my score as it is. But I won't prevent the paper from being accepted.

---

### Official Review · Reviewer_wbdk · 2023-07-21
**Lack of clarity, correctness, and relation to prior work.**

**Rating:** 5
**Confidence:** 4

**Strengths:**

(1) The data collection focused on socially significant stress-inducing tasks, gathering diverse modalities of data along with corresponding self-assessment labels.

(2) Detailed explanations were provided regarding the data construction, instruction, and maintenance methods.

(3) Limitations and ethical considerations were well addressed in the study.

**Additional Feedback:**

(1) Page 2: Regarding the limitations of existing datasets, it is mentioned that they are 'restricted in size,' but it is not specified how limited they are in terms of size.

(2) Page 2: While explaining the contributions, it is stated that the dataset is 'easy to reproduce,' but considering the need for various devices to capture respiration signals, can it genuinely be considered easy?

(3) Page 5: When describing the sensors, it is mentioned that they have a 'high signal-to-noise ratio.' The reason for this claim and relevant references need to be provided.

(4) Page 6 (Section 3.3 data annotation):
- It is recommended to state that the combinations other than those shown in Figure 3 are available in supplementary material Figure 6.
- The second paragraph starting with 'In addition, the marginal distribution~' raises doubts about its significance in the main text. It is advised to either clearly express the intended point or move it to the supplementary material.

**Clarity:**

(1) Writing should be improved.
(2) Grammar errors (e.g. Second sentence in Abstract: It is contains -> It contains)

**Correctness:**

It is constructed in a sound way. However, there are some parts where the evaluation methods and experiment design are not appropriate.

(1) In section 4 (Baselines), the mentioned baseline models are all quite outdated and not specifically developed or tailored for stress identification. If StressID is intended for stress identification purpose, it would be more appropriate to experiment with state-of-the-art or well-known models specialized for this task to demonstrate the effectiveness of this dataset.

(2) The f1-scores presented in Table 2 do not appear to facilitate a fair comparison. This is due to the varying number of tasks in each category. To enhance the fairness of the comparison, I recommend including the results of experiments for physiological and video tasks on the same 385 tasks as the audio category in the current Table 2.

**Documentation:**

Yes.

**Ethics:**

No.

**Limitations:**

Yes.

**Opportunities For Improvement:**

(1) Data omission: Since the multimodal data is not 21 hours in total (only physiological signal), it becomes challenging to claim that this dataset is the largest one. Additionally, as demonstrated in the experiments, the quantity of data significantly impacts the performance. Therefore, ensuring consistent performance by utilizing only the data containing all three modalities would be difficult.

(2)See below sections: Correctness, Clarity, Relation To Prior Work, and Additional Feedback

**Relation To Prior Work:**

Overall, yes. However, there are several unclear points that need further explanation.

(1) When describing SADVAW, it is stated, "However, it features video recordings exclusively," but it is not clear from the paper what exactly the drawbacks are when compared to StressID. More detailed elaboration on this issue is required for better understanding.

(2) The paper does not mention any drawbacks regarding the distracted driving dataset.

(3) In the "Comparison with the SOTA" subsection, it is mentioned as the first point that the approach is multi-modal, but there have been previous datasets that were also multi-modal. This claim needs to be clarified or justified.

(4) Previous research has assumed specific situations like driving because they are essential for the study. However, this aspect is not considered at all in this research, and the reason behind this decision remains unclear.

(5) Although the paper introduces StressID as the largest dataset, the number of subjects is actually smaller than the distracted driving dataset, and the exact number of samples is not compared to other studies.

**Summary And Contributions:**

This paper proposed a new dataset, StressID, which is designed specifically for stress identification. It is a multimodal dataset that contains audio, video, and respiration signal. In detail, it was collected from 65 subjects who experienced three different stimuli (stressors) while performing 11 tasks. During the data collection, the subjects conducted four self-assessments, which consisted of 4 scores: perceived stress, perceived relaxation, arousal, and valence. Based on these scores, two discrete labels were derived for model training. In total, up to 21 hours of data were collected, with video and audio data accounting for 18 hours and 7 hours, respectively, due to data-related issues. Based on the dataset, performance evaluation was conducted for both unimodal and multimodal approaches. Among them, the physiological modality showed the highest performance, possibly attributed to the largest amount of data available for this modality.

---

> ### Author Response · Authors · 2023-08-21
>
> Dear reviewer,
>
> We thank you for your very valuable feedback and comments, we have taken them into account and revised our paper accordingly. You can find answers to your comments below:
>
> #### **Opportunities For Improvement:**
> - **Missing modalities.** We are aware that the missing modalities for some participants can be a limitation. Indeed, we discuss it in Section 5. However, even so our dataset represents 19 hours of physiological data, 15 hours of video data and 6 hours of audio, amounting to close to 40 hours of annotated data in total (we apologize for our previous mistakes - after additional quality checks, we have corrected the data sizes and lengths in our revisions). Despite malfunctions during the data collection, the final dataset has recordings of respectively 65, 55, and 56 participants for the physiological, video and audio component, while competitor multimodal datasets collect data from less than 30 subjects. However, we understand why it may be fairer to qualify $\texttt{StressID}$ as *one of* the largest multimodal datasets in the related work – we have revised our paper to address this.
> #### **Correctness:**
> - **(1) Baselines.** We respectfully disagree with the reviewer’s statement. The baselines we provide are in line with the state-of-the-art in stress and emotion recognition [1-4], for the following reasons :
> 1) the importance of the extracted ECG, EDA, and respiration features in stress identification is well documented in the literature [1], and the models we have used for their classification are in line with what is done in the stress identification from physiological signals [1];
> 2) action units are used as features in state-of-the-art stress and emotion recognition systems, e.g. [2]; and the classification methodology employed in $\texttt{StressID}$ is aligned with related work on stress identification, e.g.[3]
> 3) multiple studies have highlighted the advantages of W2V models for emotion recognition from speech signals, e.g. [4]. In addition, the combination of handcrafted speech features with classical machine learning is a popular approach in speech-based stress and emotion recognition [1],[5].
>
> Nonetheless, we have revised our Section 4 to represent it more accurately, and have added some additional baselines in Section 4.1 (Table 2).
> - **(2) Comparison between unimodal and multimodal.** Thank you for this remark. As suggested, have reported additional experiments in Section 4.2 to directly compare the multimodal baseline to the unimodal ones (c.f. Table 3), by training all models on the same subset of 370 tasks featuring all modalities. We also improved the multimodal models by evaluating decision-level fusion methods. These new results are reported above in the global answer to reviewers, and highlight better the strengths of $\texttt{StressID}$.
>
> #### **Relation To Prior Work:**
> - **(1,2,4, and 5) Limitations of prior work.** Assuming specific situations, as in other existing datasets (e.g. distracted driving), is essential for some studies. However, a lack of variety in the stimuli restricts the possible applications of these datasets to specific settings. On the contrary, $\texttt{StressID}$ employs multiple stressors of varied nature, thus ensuring more versatility in potential uses. Similarly, while SADVAW is a high-quality corpus of videos, deriving applications are restricted to facial recognition exclusively, although physiological measures have been repeatedly shown to be key elements in the identification of stress. We have revised the related works (Section 2) to clarify the limitations of other existing datasets.
> - **(3) Relation of $\texttt{StressID}$ to prior work.** While there exists other multimodal datasets with experimental protocols similar to ours, such as MuSE, SWELL-KW or the distracted driving dataset, we consider each presents at least one major limitation (e.g. small number of participants for the first two, and limited usage for the last). $\texttt{StressID}$ aims to address them, and fill the gaps in the existing related work. We have revised the “Comparison with state of the art” section in the related work, hoping to highlight better the advantages of our dataset over other similar ones.
>
> #### **Additional Feedback:**
> - **(2) Ease of reproducibility.** We consider our experimental protocol is simple to reproduce in a research institute granted a minimal set-up, for the following reasons: 1) the sensors employed are accessible and relatively affordable, 2) the stressors themselves do not require any special equipment and are easy to set up, 3) we have provided a detailed description of our experimental protocol, and have additionally provided the exact instructions given to participants, in Appendix D.2.
> - **(1, 3 and 4) Other feedback.** We thank you for bringing to our attention additional opportunities for improvement. We have taken it into consideration and revised our manuscript thoroughly to make it more clear.

---

> > ### Author Response · Authors · 2023-08-21
> >
> > We hope we are able to better convey the value of our dataset, and the contributions it can bring to the advancement of stress identification. We will be happy to answer any additional questions.
> >
> > **Additional references:**
> >
> > [1] Aamir Arsalan, Syed Muhammad Anwar, and Muhammad Majid. Mental stress detection using data from wearable and non-wearable sensors: a review, 2022
> >
> > [2] Giorgos Giannakakis, Mohammad Rami Koujan, Anastasios Roussos, and Kostas Marias. Automatic stress detection evaluating models of facial action units. 2020
> >
> > [3] Mimansa Jaiswal, Cristian-Paul Bara, Yuanhang Luo, Mihai Burzo, Rada Mihalcea, and Emily Mower Provost. Muse: a multimodal dataset of stressed emotion.  2020
> >
> > [4] Li-Wei Chen and Alexander Rudnicky. Exploring wav2vec 2.0 fine tuning for improved speech emotion recognition. 2023
> >
> > [5] Naveed Ahmed, Zaher Al Aghbari, and Shini Girija. A systematic survey on multimodal emotion recognition using learning algorithms. 2023

---

> ### Comment · Reviewer_wbdk · 2023-08-30
>
> I appreciate the detailed responses and clarification. Hence, I would change the rating from 4 to 5.

---

### Official Review · Reviewer_9RXE · 2023-07-21
**StressID: a Multimodal Dataset for Stress Identification**

**Rating:** 6
**Confidence:** 4
**Clarity:** The paper is well written and clearly…

**Strengths:**

The contribution of this work will help in the building of robust and reliable, frameworks for stress recognition. The multimodal nature of the data is a strength. Further, the authors include an easy to reproduce experimental protocol such that more data could be collected in the future to augment this data set. The authors have also thought through how the data can be used and demonstrate some examples of stress detection models using this data in practice.

One major strength is the combination of the two types of labels: the stimulus label, as well as the feelings self-reports. The seven interactive stressors seem to be well validated methods that are based in the literature, which is a strength.

**Additional Feedback:**

As one of the problems discovered was that the verbal tasks are associated with higher levels of stress, could the team include suggestions for their own or other groups future data collection to include a non-stressful verbal task, like talking on the phone with a friend?

**Correctness:**

As far as the reviewer can tell, not being an expert in the field of stress, the paper seems correct.

**Documentation:**

yes

**Ethics:**

No. It would be important to report the compensation structure, and whether there was any incentive for people to be more likely to share their data publicly. The credentialized access helps to protect the data such that access is more likely to be given to appropriate and well-intentioned users.

**Limitations:**

The limitations are well described with the exception of a few noted problems, which are explained above in the opportunities for improvement.

**Opportunities For Improvement:**

The largest limitation in this data set is the extreme gender imbalance (18 women and 47 men) after the data, cleaning steps, it is not clear how many of the data sets are usable among women and men (and overall demographics of the usable data), and that would be critical to report.

At the end of the third block of tasks, the fact that participants are asked to designate the task perceived, as most stressful probably will suffer from recency bias. The author should explore whether those tasks that occur or later are more likely to be rated as more stressful.

The study participant compensation is not described. It is unclear whether they were compensated at all, and whether they were compensated differently for choosing the data sharing options a or B. Also, it should be reported how many of the people choosing option a or B or women or men.

Was time of day recorded, and if not, could that be an important factor affecting the feelings of the participants?

Given that the study was conducted in France, but the experiments were conducted entirely in English, did that study set up limit the types of demographics that could participate in the study? What might be the effects of that on the data itself?

In the binary labeling, would it potentially make sense to remove ratings that are toward the middle of the ratings? Is there enough data at the extremes to do this to improve the stress detection algorithm? What would be the pros and cons of this?

It is not clear why the multimodal baseline model underperforms as compared to the other models. Please give some insight here.

Please define what action units are in the video data.

**Relation To Prior Work:**

Yes. However, there are a few datasets that are similar and the differences between their work and these data are described in the second paragraph under Related Work. However, there should be clear about the types of stress inducing, stimuli that those other studies did as compared with the stimuli in this data set. It is not clear what the literature basis of the stimuli are or how well studied they are outside of these data sets.

**Summary And Contributions:**

StressID is a new multimodal dataset for stress detection. It contains facial expression video, audio, and physiological signal recordings, labels of the stimuli, and self-reported ratings of feelings associated with the different stimuli. The major contributions of this work include that not many multi-modal datasets exist, and this is roughly two times larger than the others and also contains more of a variety of stimuli. While the novelty isn’t huge, it is still an important contribution to the field that will benefit future researchers.

---

> ### Author Response · Authors · 2023-08-21
>
> Dear reviewer,
>
> Thank you for your feedback and very relevant comments. We would like to thank you for recognizing the value of our dataset and the contributions it can bring to the field of stress recognition. You can find answers to your questions and comments below:
>
> #### **Opportunities For Improvement:**
> - **Gender imbalance.** We acknowledge the gender imbalance in our dataset can lead to bias in deriving analyses. We bring it to the attention of the readers by discussing the ethical considerations of working with human data in Section 6. An extended discussion is also included in Appendix 2. ;
> - **Potential bias.** We have concluded from the answers to the survey question that the experiment is not subject to recency bias, Counting2 is most commonly designated as the most stressful task among participants, while it occurs relatively early in the experiment. This analysis is reported in Appendix E.1.
> - **Participants compensation.** All subjects were recruited on a voluntary basis, i.e. without compensation. We acknowledge this information might not have been clear enough in our first version. Therefore, we have modified Section 3.2.1 to mention it explicitly. We have also added information on the gender of participants choosing option B, i.e. two women and one man.
> - **Time of the day.** Although recorded we have not reported time of the day in the paper. We do not believe this has a significant impact on our dataset since the aim of $\texttt{StressID}$ is to record responses to stress inducing stimuli – being therefore more focused on short-term stress.  Moreover, by making sure the recording conditions and experimental set-up of $\texttt{StressID}$ were identical to all participants, and by starting the experiment with a guided breathing exercise intended for resetting their affective states to neutral, we minimize, as much as possible, any variability due to external factors susceptible to impact the mental state of participants.
> - **Participants recruitment.** Subjects for $\texttt{StressID}$ were recruited by email, and word of mouth primarily. The vast majority of people interested in it were proficient in English – being students/workers in a very international environment. For this reason, we do not consider this aspect to have restricted the demographics of the study. However, we acknowledge this might have had effects on the study, as speaking English might have been a stressor itself for non-native speakers. We thank you for pointing this out, we have accordingly revised the dataset Limitations (see Section 5).
> - **Labels.** It is, indeed,  possible to focus on extreme cases only, as the self-ratings are rather coherently distributed across the dataset (Figure 3 in the paper). An advantage of doing so would be the possibility to focus on specific types of emotional response – a good example of application being an anomaly detection algorithm trained to detect cases of acute stress only. However, for systems developed to discriminate between different types of responses, e.g. positive stress vs. negative stress, having access to a larger range of self-assessments is essential, and restricting the dataset to low/high values of stress could lead to a significant loss of information. The labels we presentare designed for an example of use case. However, we provide the original unprocessed self-reportings of the participants in order to allow users of $\texttt{StressID}$ to design their own labels, best suited for their applications (as, for instance, the analysis of extreme cases)..
> - **Multimodal baseline.** In our initial results, the multimodal model performed worse than unimodal ones as it was trained on a significantly smaller subset. We acknowledge the baseline section did not accurately reflect the advantages of $\texttt{StressID}$. Therefore, we revised Section 4, and  have included additional experiments (Section 4.2) to directly compare the multimodal baseline with the unimodal ones (Table 3), by training all models on the same subset of 370 tasks featuring all modalities. These new results are reported above in the global answer to reviewers, and highlight better the strengths of multimodal learning.
> - **Action Units.** AUs are fine-grained facial muscle movements - extracted from face images - that correspond to the fundamental actions of individual muscles or groups of muscles. We have revised the video baseline description in Section 4.1., to make it clearer to the readers. Additionally, we have included  a visual example of extracted AUs in Appendix F.1.
>
> #### **Relation To Prior Work:**
> - **Stressors.** The stress-inducing stimuli used in $\texttt{StressID}$ (and other similar datasets[1]), are based on well-established clinical experimental stress studies, developed on independent studies (i.e. see [2] for an extensive review) not related to the datasets. We refer to these studies in Section 3.1.1.

---

> > ### Author Response · Authors · 2023-08-21
> >
> > We hope to have clarified those points for you, and would be happy to answer any additional questions.
> >
> >
> > **Additional references :**
> >
> > [1] Aamir Arsalan, Syed Muhammad Anwar, and Muhammad Majid. Mental stress detection using data from wearable and non-wearable sensors: a review. 2022
> >
> > [2] Anjana Bali and Amteshwar Singh Jaggi. Clinical experimental stress studies: methods and assessment.2015.

---

> > ### Comment · Reviewer_9RXE · 2023-08-28
> > **Inherent flaw in the data collection process would lead to bias**
> >
> > The responses on the gender imbalance and the recruitment process have uncovered that there was a lack of appropriate planning for recruiting an equitable and balanced study population, which is critical for generating a dataset that will minimize the risk of bias in algorithm development. This unfortunately reduces the reviewer's enthusiasm for this work.

---

> > > ### Author Response · Authors · 2023-08-29
> > > **Precisions on participants recruitment and additional experiments**
> > >
> > > We thank you for your additional feedback.
> > >
> > > We agree a balanced study is crucial for developing bias-free analyses. We have therefore added new experiments in Appendix F of the supplementary material to: 1) sensitize users to the effect of gender bias; 2) demonstrate how $\texttt{StressID}$ can effectively be used to build equitable applications. We address your points in more detail below:
> > >
> > > - **Recruitment process:** Our project has been developed and conducted within a defined scope, and as a consequence our recruitment has been strongly focused on students and workers of STEM environments, similarly to [1],[2],[3],[5]. In addition, gender imbalance is a limitation we share with competitor datasets [1],[2],[3],[4],[5] (with the proportion of female participants going as low as 20% of the dataset in some cases), and a common issue of human data collection more generally[6],[7]. Despite numerous email campaigns, and our multiplied efforts across student campuses, research laboratories and corresponding administrative offices to recruit a balanced population, the largest part of the participants remained male. We do not consider this imbalance to be a consequence of a lack of planning, but rather representative of the female/male ratio in STEM studies and workforce [8].
> > > - **Consequence of imbalance in algorithm development :** To increase users' awareness to this issue, we have reported additional experiment in Appendix F.2.3. In order to investigate the possible effect of gender imbalance, we studied the classification of stress on a subset of $\texttt{StressID}$ composed of 18 female and 18 randomly selected male participants. These new results highlight the importance of balance in a dataset, and simultaneously demonstrate the possibility of developing algorithms achieving good classification performances on balanced subsets of $\texttt{StressID}$. We aim to be as transparent as possible about this aspect, making it clear that researchers need to be aware of potential representation bias in their analyses. We therefore encourage them to anticipate the possible consequences and build equitable systems, as shown possible by our additional experiments .
> > >
> > > The field of stress identification lacks multimodal datasets large enough to foster the development of varied and equitable algorithms. $\texttt{StressID}$ can help further the research on stress recognition by answering these needs. For instance, the gender-balanced subset of $\texttt{StressID}$ still represents 36 participants, while competitor similar multimodal datasets collect data from less than 30 subjects in total, rendering such analyses impossible.
> > >
> > > **Additional references:**
> > >
> > > [1] Koldijk et al., The swell knowledge work dataset for stress and user modeling research. 2014.
> > >
> > > [2] Markova et al., Clas: a database for cognitive load, affect and stress recognition. 2019
> > >
> > > [3] Schmidt et al. ,Introducing wesad, a multimodal dataset for wearable stress and affect detection. 2018
> > >
> > > [4] Taamneh et al., A multimodal dataset for various forms of distracted driving. 2017
> > >
> > > [5] Jaiswal et al., Muse: a multimodal dataset of stressed emotion. 2020.
> > >
> > > [6] D'Mello et al., Exclusion of females in autism research: Empirical evidence for a "leaky" recruitment-to-research pipeline. 2022
> > >
> > > [7] Pinho-Gomes et al., Dementia clinical trials over the past decade: are women fairly represented? 2022
> > >
> > > [8] UNESCO. Women in Science: Fact Sheet No. 55. FS/2019/SCI/55. June 2019

---

### Author Response · Authors · 2023-06-16
**Private secured link for accessing the StressID dataset**

Dear reviewers,

First, we would like to thank you for taking the time to review our work.

Following our ethical protocol, and given the sensitive nature of our data (i.e. facial videos), our dataset is made accessible through credentialized access only.

All researchers can request access to the data following the procedure on the $\texttt{StressID}$ webpage : https://project.inria.fr/stressid/

We guarantee that the established credentialization procedure in place is:
- open to a large section of the public, i.e. researchers
- provides rapid response and access to the data
- is guaranteed to be maintained on the long-term.

Information regarding hosting and maintenance can be found in the supplementary material.

Please find below a secured link to the data and 3 credentials, specifically put in place for the review process.

https://repo-sam.inria.fr/StressID/

- reviewer1 QJHXBQuhrW42NYQ8
- reviewer2 jV5UyBvXSiwK1K6Q
- reviewer3 kL1jziguHqBp7Qff

We kindly remind you that information shared here is confidential, you may not grant anyone else access to the dataset by sharing the credentials, your access and use is related to the review process only, that it is performed through institutional computers compliant with the security policy of your establishment (up-to-date antivirus software, etc.). Please read the license agreement in the repository if you download the dataset.

Thank you for your time and consideration.

Best regards.

---

> ### Comment · Area_Chair_79iX · 2023-07-04
> **Query about credentialised access**
>
> Dear authors,
> We have 5 reviewers assigned to this paper with the following IDs:
> 1. QzsT
> 2. jqtB
> 3. qPpi
> 4. wbdk
>
> The reviewers do not know their ID as a number at the point. Is there an issue if someone accesses a dataset with one set of credentials and then another one later? Not sure if you are logging IP addresses with the credentials.
>
> Best wishes
> Your AC

---

> > ### Author Response · Authors · 2023-07-04
> > **Query about credentialised access**
> >
> > Dear AC,
> >
> > There is no problem in doing so.
> >
> > Thanks for asking,
> >
> > Best regards.

---

> > > ### Comment · Area_Chair_79iX · 2023-07-05
> > >
> > > Dear Authors,
> > > There seems to be an issue with accessing your dataset however. Here is the issue from one of the reviewers:
> > >
> > > "Unfortunately, I cannot access the dataset using the provided account and password from the author; my browser shows a 500 error. I am not sure if it is an issue with server-side or browser settings."
> > >
> > > Best wishes,
> > > your AC

---

> > > > ### Author Response · Authors · 2023-07-05
> > > >
> > > > Dear reviewer,
> > > > We are sorry to hear that.
> > > > We have done some tests using your credentials on chrome, firefox and safari, with different wi-fi and different locations. It worked for us.
> > > >
> > > > Could you provide us some more details about your environment so we try to reproduce it?
> > > > We are in contact with our technical office, who is providing us support on this.

---

> > > > > ### Author Response · Authors · 2023-07-05
> > > > >
> > > > > Dear reviewer,
> > > > >
> > > > > our technical office suggested to double check the url (https://repo-sam.inria.fr/StressID/) and the credentials.
> > > > > Then he asked if you can try again and note down the exact time (with minutes) at which the error shows up, so he can better investigate the log.
> > > > >
> > > > > Thanks.

---

> > > > > > ### Comment · Area_Chair_79iX · 2023-07-07
> > > > > >
> > > > > > Not sure if you can see the reviewer response: the dataset access issue is now resolved. Thanks.
> > > > > > Your AC

---

### Author Response · Authors · 2023-08-21

We would like to thank the reviewers for their valuable feedback and the interest shown in our work.  We would also like to thank them for acknowledging the quality of $\texttt{StressID}$, its collection process, and recognizing the contribution our dataset can bring to building robust and reliable frameworks for stress identification. Indeed, our aim is to further the research on human stress recognition and improve the understanding of the relationships between physiological and behavioral responses by proposing a multimodal dataset designed for stress identification. In this respect, our work does not pretend to make model contributions, but to provide several baselines to illustrate the predictive potential of our dataset, and to be used as a comparison point for future researchers


The main concerns raised by the reviewers concern the positioning of our work with other previous related works and the ability of our baselines to reflect the advantages of $\texttt{StressID}$. We have revised our paper to address these issues. Our major modifications are summarized below :

- We have revised the “Comparison with state of the art” paragraph in the related work (Section 2). The new version highlights better the advantages of our dataset over other similar ones: while there exist other multimodal datasets similar to ours, such as MuSE, SWELL-KW, or the distracted driving dataset, we consider each presents at least one major limitation (e.g. small number of participants for MuSE and SWELL-KW, and limited usage for the distracted driving dataset). $\texttt{StressID}$ aims to address these limitations, and fills the gaps in the existing previous work; indeed $\texttt{StressID}$ is the first multimodal dataset (including both physiological and behavioral modalities) for stress identification that is recorded on a large number of participants – and that also features a wide range of stimuli ensuring the versatility of deriving applications.

- We have included more baselines and improved the multimodal baselines by adding models employing feature level and decision level fusion in Section 4. We have reported additional experiments in section 4.2 to directly compare the multimodal baseline to the unimodal ones, by training all models on the same subset of 370 tasks featuring all modalities. Additional results for all other modality combinations are reported in the supplementary material. These new results reflect the advantages of $\texttt{StressID}$. The table below summarizes the new results (Table 3 in the paper):


$$
\\begin{array}
        \\hline
        &\\textbf{2-class} &  & \\textbf{3-class} & \\\\
        \\textbf{Baseline} & F1-score & Accuracy & F1-score & Accuracy \\\\
        \\hline
        \\text{Physiological} & 0.66 \\pm 0.05 & 0.58 \\pm 0.04 & 0.50 \\pm 0.05 & 0.48 \\pm 0.06 \\\\
        \\text{Video} & 0.67 \\pm 0.03 & 0.62 \\pm 0.04 & 0.58 \\pm 0.05 & 0.56 \\pm 0.05\\\\
        \\text{Audio} & 0.67 \\pm 0.04 & 0.62 \\pm 0.04 & 0.56 \\pm 0.06 & 0.54 \\pm 0.06 \\\\
        \\hline
         \\text{Feature fusion + SVM} & 0.64 \\pm 0.09 & 0.56 \\pm 0.05 & 0.55 \\pm 0.06 & 0.51 \\pm 0.05\\\\
         \\text{Feature fusion + MLP} & 0.66 \\pm 0.04 & 0.61 \\pm 0.03 & 0.51 \\pm 0.07 & 0.51 \\pm 0.07\\\\
         \\text{Feature fusion + DBN} & 0.58 \\pm 0.06 & 0.52 \\pm 0.05 & 0.30 \\pm 0.09 & 0.32 \\pm 0.04\\\\
        \\hline
         \\text{SVM + Sum rule fusion} & \\textbf{0.72 $\\pm$ 0.05} & 0.64 \\pm 0.05 & 0.62 \\pm 0.05 & \\textbf{0.58 $\\pm$ 0.07}\\\\
         \\text{SVM + Product rule fusion} & 0.71 \\pm 0.05 & 0.63 \\pm 0.05 & 0.61 \\pm 0.05 & 0.56 \\pm 0.07\\\\
        \\textbf{SVM + Average rule fusion} & \\textbf{0.72 $\\pm$ 0.05} & \\textbf{0.65 $\\pm$ 0.05} & \\textbf{0.63 $\\pm$ 0.05} & \\textbf{0.58 $\\pm$ 0.07}\\\\
         \\text{SVM + Maximum rule fusion} & \\textbf{0.72 $\\pm$ 0.05} & 0.64 \\pm 0.05 & 0.61 \\pm 0.06 & 0.57 \\pm 0.07\\\\
        \\hline
\\end{array}
$$



In addition to these major remarks, in the following, we address other comments and questions hoping to clarify remarks raised by each reviewer.

---

### Author Response · Authors · 2023-08-28

Dear reviewers,

Since the author/reviewer discussion period is approaching the deadline, we wonder if you have any remaining questions or concerns after our response. We are happy to provide more information.

Thank you.

---

### Decision · Program_Chairs · 2023-09-22

**Decision:**

Accept (Poster)

**Comment:**

This paper provides a new multimodal dataset for the important and challenging area of stress identification. The reviews were mixed in terms of whether the paper was of sufficient quality. The set of tasks used for providing stress are novel and interesting, especially given the context of the population. In light of the overall comments and understanding the important context in which the dataset was collected, I have decided to recommend acceptance.

The reviewers expressed doubt about the novelty of the dataset given the evidence from Table 2 of the paper.
The rewrite of the "comparison of the state of the art" section is still unfortunately not clear e.g. "Indeed, StressID employs diverse and carefully selected tasks, and rather than relying on task-based ground truths,...". However, given the confidence of other authors and also the discussion I make below, I think the work should be accepted providing the changes below are made to the presentation of the work.

Regarding the gender imbalance as highlighted by reviewer 9RXE. If the aim of the dataset is to train models that can learn stress, anything learned from this dataset could be extremely biased towards males. There is some arguments given by the authors related to the origins of the project funding being about students and workers in STEM environments. This is fine. However, it is not reflected obviously back in the paper presentation at the moment and I think this is problematic for the spirit of the datasets and benchmarks track. It is important to highlight novel application specific demands that inspire the collection of new datasets.

I think that designing the dataset and presenting the dataset to focus on the goals of the project more directly would make the paper stronger. As it stands now the motivation of the design of the stimuli and how it differs from related datasets is rather confusing. If there are concerns about the dataset being too specific, there are always opportunities to specify in the writing of the paper how this dataset might still be useful in a more generalised context.

Finally, please address in presentation of the paper either in the main text or supplementary: e.g. what new research questions are enabled by the data wrt the ML community, application domain, as well as for affect understanding?